# Learning Retrieval Models with Sparse Autoencoders

**Thibault Formal**[*]
Naver Labs Europe
thibault.formal@gmail.com

**Maxime Louis**[*]
Naver Labs Europe
maxime.louis.x2012@gmail.com

**Hervé Déjean**
Naver Labs Europe
herve.dejean@naverlabs.com

**Stéphane Clinchant**
Naver Labs Europe
stephane.clinchant@naverlabs.com

## Abstract

Sparse autoencoders (SAEs) provide a powerful mechanism for decomposing the dense representations produced by Large Language Models (LLMs) into interpretable latent features. We posit that SAEs constitute a natural foundation for Learned Sparse Retrieval (LSR), whose objective is to encode queries and documents into high-dimensional sparse representations optimized for efficient retrieval. In contrast to existing LSR approaches that project input sequences into the vocabulary space, SAE-based representations offer the potential to produce more semantically structured, expressive, and language-agnostic features. Building on this insight, we introduce SPLARE, a method to train SAE-based LSR models. Our experiments, relying on recently released open-source SAEs, demonstrate that this technique consistently outperforms vocabulary-based LSR in multilingual and out-of-domain settings. We developed SPLARE-7B, a multilingual retrieval model capable of producing generalizable sparse latent embeddings for a wide range of languages and domains, achieving top results on MMTEB's multilingual and English retrieval tasks. We also propose a 2B-parameter variant with a significantly lighter footprint.

## 1 Introduction

Embedding models have become a pivotal tool for search systems, enabling the better capture of semantic relationships between queries and documents across various domains and modalities. This trend has been further accelerated by the advent of Retrieval-Augmented Generation (RAG) Lewis et al. (2020) and agent-based systems, which impose even higher demands on retrieval performance and robustness. Recently, dense embedding models Reimers & Gurevych (2019); Karpukhin et al. (2020), which map inputs into single dense vectors, have demonstrated impressive performance on the (M)MTEB benchmark (Muennighoff et al., 2023; Enevoldsen et al., 2025). Specifically, embedding models relying on large (V)LLM backbones have become the de facto approach for generalist multilingual Lee et al. (2025b); Zhang et al. (2025); Wang et al. (2024a); Lee et al. (2025a); Li et al. (2023b) or even multi-modal models Günther et al. (2025); Faysse et al. (2025); Xu et al. (2025a)—marking a shift away from encoder-only language models which have defined the state of the art for years (Izacard et al., 2022; Karpukhin et al., 2020; Xiong et al., 2020).

Learned Sparse Retrieval (LSR) methods Formal et al. (2021); Mallia et al. (2021); Nguyen et al. (2023); Kong et al. (2023) have achieved state-of-the-art performance on widely used English-centric benchmarks Thakur et al. (2021); Bajaj et al. (2018); Craswell et al. (2021) and have demonstrated strong generalization when compared to dense embedding models (Formal et al., 2022b; Lupart et al., 2023; Déjean et al., 2023). Beyond their efficiency, these approaches provide a level of interpretability that is particularly valuable in production systems. Models such as SPLADE Formal et al. (2021; 2022a); Lassance et al. (2024) operationalizex this idea by representing documents

---

[*]Contributed equally. Corresponding authors.

and queries as sparse, weighted bag-of-words over the vocabulary space of their backbone model. While originally developed for encoder-only architectures such as BERT Devlin et al. (2019), recent work has explored adapting SPLADE to LLM backbones (Qiao et al., 2025; Doshi et al., 2024; Xu et al., 2025b; Zeng et al., 2025; Soares et al., 2023; Ma et al., 2025). However, these models remain limited to English-centric contexts and struggle to match state-of-the-art performance on more comprehensive benchmarks like MMTEB which place greater emphasis on generalization across novel domains and languages. Unlike dense retrieval, which models relevance within a continuous embedding space, LSR methods are inherently constrained by the fixed vocabulary of their underlying backbone, which incurs issues such as tokenization redundancy (Lei et al., 2025). This limitation also makes it significantly harder to handle multilingual or cross-lingual retrieval Nair et al. (2023; 2022); Lassance (2023)—and even more so when extending to multimodal settings (Nguyen et al., 2024). We hypothesize that this is a key reason why LSR models have recently fallen behind dense approaches[1].

In the context of LLMs, Sparse Autoencoders (SAEs) Makhzani & Frey (2013); Huben et al. (2024); Bricken et al. (2023) decompose dense token representations into sparse vectors of latent features. These features have been shown to exhibit desirable properties: they are largely mono-semantic (most features correspond to a single interpretable concept), multilingual (remaining largely language-agnostic), and even multimodal (generalizing across modalities in multimodal LLMs) (Bricken et al., 2023; Templeton et al., 2024; Lieberum et al., 2024; Huben et al., 2024; He et al., 2024; Deng et al., 2025). While SAEs have generated significant excitement for mechanistic interpretability, recent work has also highlighted their limitations, showing that they can struggle to transfer effectively to certain downstream tasks (Kantamneni et al., 2025; Smith et al., 2025).

In this work, we argue and empirically demonstrate that SAEs are a natural fit for LSR models: their learned latent features provide a semantically-grounded representation space for sparse retrieval which is particularly advantageous in domains or languages where vocabulary-based approaches may underperform. To this end, we propose a new LSR approach that represents queries and documents as sparse vectors over a latent vocabulary space, by replacing the standard language modeling (LM) head with pre-trained SAEs such as Llama Scope (He et al., 2024). More specifically, our contributions are as follows:

- We introduce SPLARE—for SParse LAtent REtrieval—a new LSR approach relying on pre-trained SAEs;
- We conduct a systematic investigation of the advantages of using a latent vocabulary—compared to the standard LLM vocabulary—across a comprehensive set of benchmarks spanning diverse tasks, domains, and languages;
- Finally, we introduce a new 7B multilingual latent sparse retriever that supports 100+ languages through cross-lingual transfer and achieves competitive results on the MMTEB *retrieval* benchmark. SPLARE is the first LSR model to rival state-of-the-art dense approaches on MMTEB, *by fine-tuning only on a large open-source dataset, without additional pretraining or data augmentation.* We additionally develop a compact and efficient 2B counterpart.

## 2 BACKGROUND

We first provide some background on Sparse Autoencoders as well as Learned Sparse Retrieval. SPLARE can be understood as synthesizing these two research directions into a unified framework.

### 2.1 SPARSE AUTOENCODERS

Given activations $x \in \mathbb{R}^d$ from a language model, a Sparse Autoencoder (SAE) is a single hidden layer model, comprising an encoder and a decoder:

$$z = f(\boldsymbol{W}_{\text{enc}}x + \boldsymbol{b}_{\text{enc}}), \quad \hat{x} = \boldsymbol{W}_{\text{dec}}z + \boldsymbol{b}_{\text{dec}} \qquad (1)$$

where $z \in \mathbb{R}^{|\mathcal{W}|}$, with $|\mathcal{W}| >> d$ corresponding to the width of SAE, i.e., the number of features in the latent space. SAEs, as a class of autoencoders, are trained using a standard reconstruction

---

[1]As of the time of writing, no sparse retrieval model is listed on the MTEB(Multilingual, v2) leaderboard.

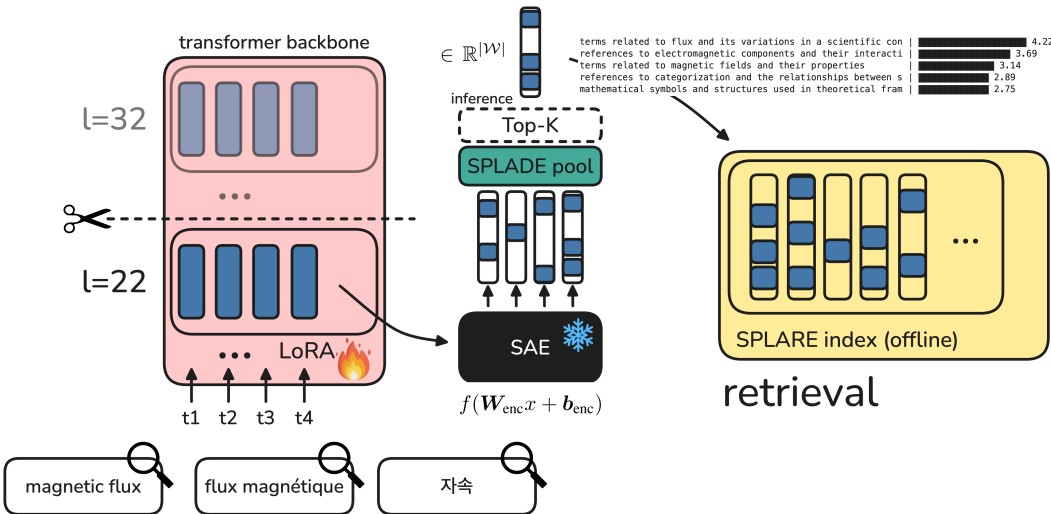

Figure 1: Overview of SPLARE. A pre-trained SAE can be inserted at any layer $l$ of the LLM to get sparse latent representations of input tokens. These token-level representations are then aggregated into a single sparse vector using a pooling mechanism analogous to SPLADE. During training, we only fine-tune the LLM parameters (via LoRA adapters) while keeping the SAE frozen. We show the top-5 activated features Lin (2023) for the English query—for our final SPLARE model.

objective $\mathcal{L} = \|\hat{x} - x\|^2$. Sparsity in the decomposition is induced through suitable activation functions $f$ such as ReLU Bricken et al. (2023), Top-K Makhzani & Frey (2013); Gao et al. (2025) or JumpReLU Rajamanoharan et al. (2024), and/or regularization penalties such as $\ell_1$. Several works have demonstrated that SAEs can recover highly monosemantic features, many of which are language-agnostic—responding consistently to the same concepts across languages—and, in some cases, even multimodal (Huben et al., 2024; Bricken et al., 2023; Templeton et al., 2024; Lieberum et al., 2024; He et al., 2024; Cunningham & Conerly, 2024; Deng et al., 2025). Large Sparse Autoencoders are also notoriously hard and costly to train. Recently, high-quality large scale open-source SAEs have become available to the research community. In particular, we rely in this work on the Llama Scope series of models He et al. (2024) which offers SAEs trained on Llama-3.1-8B and the Gemma Scope suite Lieberum et al. (2024) which offers SAEs trained on Gemma-2-2B, 9B and 27B models.

## 2.2 SPLADE

Learned Sparse Retrieval (LSR) models aim to map input sequences into high-dimensional sparse representations for efficient retrieval. Among these approaches, the SPLADE family of approaches Formal et al. (2021; 2022a); Lassance et al. (2024) has emerged as the state-of-the-art method, achieving performance comparable to or exceeding that of dense embedding models in many settings. Given an input sequence tokenized as $t = (t_1, t_2, \ldots, t_n)$ and fed through all the layers of the transformer, SPLADE generates a sequence of logits $(v_1, v_2, \ldots, v_n)$ by projecting each final hidden state $(h_1, h_2, \ldots, h_n)$ onto the vocabulary space $\mathcal{V}$ using the language modeling head. The weights $(v_{ij})_{j \in \mathcal{V}}$ correspond to an unnormalized log-probability distribution over $\mathcal{V}$ for token $t_i$, where each output dimension $j$ is actually associated with the token it represents. To obtain a single sequence-level representation, SPLADE first applies a term saturation function, before max-pooling over the sequence:

$$u_j = \max_{i=1\ldots n} \log\left(1 + \text{ReLU}(v_{ij})\right), j \in \mathcal{V} \tag{2}$$

Given these sparse representations $u \in \mathbb{R}^{|\mathcal{V}|}$ for queries and documents, relevance scores are computed as a sparse dot product $s(q, d) = <u^q, u^d>$. This operation can be efficiently supported using inverted index structures together with specialized query processing techniques (Tonellotto et al., 2018; Bruch et al., 2024c; Zobel & Moffat, 2006).

## 3 METHOD

### 3.1 SPLARE

Conceptually, SPLARE closely parallels SPLADE but operates in the latent representation space. Rather than projecting the final hidden states of the language model onto the vocabulary space via the LM head, SPLARE employs sparse autoencoders to transform representations from a selected layer into a sparse latent space, which can be interpreted as a latent vocabulary.

Let $(\boldsymbol{W}_{\mathrm{enc}}, \boldsymbol{b}_{\mathrm{enc}})$ in Eq. 1 denote the SAE's encoder parameters at a given layer $l$ of the transformer[2]. Similarly to SPLADE, we can obtain sequences of sparse latent logits $(w_1, w_2, \ldots, w_n)$ by mapping the hidden states at layer $l$ with the SAE encoder as illustrated in Figure 1. The weights $(w_{ij})_{j \in \mathcal{W}} \in \mathbb{R}^{|\mathcal{W}|}$ contain the sparse list of latent features associated with token $i$ in the input sequence. It can be used in place of the vocabulary decomposition to compute sequence-level representations for input queries or documents into a sparse set of latent features using the same type of pooling mechanism as in Eq. 2—which we refer to as SPLADE-pool in Figure 1.

### 3.2 TRAINING

**Training LSR Models.** The training procedure for LSR models mirrors that of dense embedding models. While contrastive learning Oord et al. (2018); Chen et al. (2020) is the de-facto approach to train state-of-the-art dense models Lee et al. (2025b); Zhang et al. (2025), we instead adopt a distillation-based approach using a cross-encoder teacher model Nogueira & Cho (2020) to train our sparse embeddings. Specifically, we optimize the Kullback–Leibler divergence between the teacher and student relevance distributions (Lin et al., 2020). Given a query $q$, $(d_1, d_2, \ldots d_m)$ which contains a positive document and a pool of hard negatives, $(s_1, s_2, \ldots s_m)$ the corresponding teacher scores for documents $d_i$ with respect to $q$, and $\tau$ a temperature parameter, the training loss is given by:

$$\mathcal{L}_{\mathrm{KL}} = \sum_{i=1}^{m} p_i \left(\log p_i - \log \hat{p}_i\right), \quad \hat{p}_i = \frac{e^{s(q,d_i)/\tau}}{\sum_j e^{s(q,d_j)/\tau}}, \quad p_i = \frac{e^{s_i}}{\sum_j e^{s_j}} \qquad (3)$$

**Distillation *vs* Contrastive Learning.** Distillation is a common toolbox to train retrieval models Hofstätter et al. (2020); Lin et al. (2020), but has been overlooked in the context of LLM-based embeddings. State-of-the-art embedding models typically rely on contrastive learning, which is generally effective but suffers from well-known issues such as false negatives. As a result, many recent systems incorporate filtering mechanisms for negative samples—often using cross-encoders or LLMs—which can themselves be viewed as implicit forms of distillation (Lee et al., 2025b; de Souza P. Moreira et al., 2025). In addition, since our retrieval model mirrors SPLADE (with the vocabulary projection being the only difference), we follow the established training practices for SPLADE (Formal et al., 2022a; Lassance et al., 2024).

**Sparsity.** To encourage sparsity in query and document representations, LSR models are typically trained with a sparsity-inducing regularization term, analogous to that used in SAEs. We use the FLOPS loss Paria et al. (2020) employed in SPLADE. The final loss is:

$$\mathcal{L} = \mathcal{L}_{\mathrm{KL}} + \lambda_q \ell_{\mathrm{FLOPS}}^q + \lambda_d \ell_{\mathrm{FLOPS}}^d \qquad (4)$$

The sparsity of LSR approaches plays a crucial role in determining both effectiveness and computational efficiency on retrieval benchmarks. However, the sparsity induced by $\mathcal{L}$ can vary significantly depending on the model configuration, backbone architecture, SAE suite, and dataset characteristics. Achieving a desired target sparsity would require continuous adjustment of $\lambda_{d,q}$. To mitigate this challenge and establish a more robust training setup, we additionally apply Top-K pooling *at inference time*, as illustrated in Figure 1. This strategy allows us to train a single model with moderate sparsity—using fixed, conservative values of $\lambda_{d,q}$—while systematically studying the effect of pooling without the need for re-training. Although some prior works have entirely replaced explicit

---

[2]Note that we only rely on the encoder parameters, as we only aim to extract sparse features from representations. Also note that we consider SAEs trained on the residual streams of the transformer.

sparsity regularization with Top-K pooling (Lassance et al., 2023; Doshi et al., 2024), our initial experiments with this approach yielded inferior results. Note that Top-K acts as a strict upper bound: depending on the dataset, queries and documents may contain fewer active dimensions due to the inherent sparsity of the representations.

Finally, we note that while SPLARE is initialized with an SAE—which produces sparse token-level representations—sequence-level sparsity at initialization remains relatively high (e.g., a few thousands non-zero values). As a result, additional sparsity regularization is required to ensure the model achieves the desired efficiency. It is also worth noting that LSR models are usually hard to train and require a careful initialization of the projection head. While the LM head or a SAE can provide a suitable initialization, training an LSR model entirely from scratch is highly difficult and consistently results in much lower performance—when converging.

## 4 EXPERIMENTAL SETUP

**Training Data.**  We conduct two large sets of experiments: § 5 contains various ablations and analyses for models trained on English data on the MS MARCO dataset (Bajaj et al., 2018). In § 6, we further extend training to a larger set of publicly available data, including multilingual datasets. We do not prepend any special instructions or prefix to our input sequences—which could only likely yield further improvements. To ease reproducibility, we also refrain from any form of pre-finetuning or synthetic data generation Lee et al. (2025b); Günther et al. (2025); Zhang et al. (2025), both of which have recently become common practice for achieving top results on the MTEB benchmark. We detail in Appendix A our two training settings.

**Evaluation.**  MTEB Muennighoff et al. (2023) and MMTEB Enevoldsen et al. (2025) are the most widely adopted benchmarks for evaluating embedding models. Our evaluation focuses only on the *retrieval* subsets of these benchmarks, excluding other task categories. In addition to the English and Multilingual splits, we also report results on domain-specific subsets of MTEB, including Code, Medical, Law, and Chemical domains. Given SPLARE's strong performance in multilingual settings, we further place particular emphasis on this aspect by including language-specific splits of MMTEB for five languages, as well as evaluations on the MIRACL Zhang et al. (2023) and XTREME-UP Ruder et al. (2023) datasets. The latter introduces a challenging cross-lingual retrieval task, requiring retrieval from an English corpus using queries from low-resource languages. We also report results on MS MARCO Bajaj et al. (2018) and BEIR Thakur et al. (2021) (Appendix C).

While our approach is broadly applicable to any pre-trained SAE, we conduct the majority of our experiments using the Llama Scope model suite He et al. (2024), built on Llama-3.1-8B (et al., 2024). During training, we fine-tune the backbone with LoRA adapters Hu et al. (2022) while keeping SAE parameters frozen. Preliminary experiments indicated that this strategy not only improves performance but also simplifies training. Moreover, it preserves the interpretability of the latent feature space (Lin, 2023). As in prior work Zeng et al. (2025); BehnamGhader et al. (2024); Lei et al. (2025), we enable bidirectional attention across all backbones and pretrain them with Masked Next Token Prediction. Following the exact procedure of Zeng et al. (2025), we mask 20% of tokens in the MS MARCO corpus and train for $10k$ steps which takes about five hours. Bidirectional attention is particularly important for LSR models since pooling occurs at every position of the input sequence, unlike dense models that rely on the <EOS> token. Full details of our experimental hyperparameters are provided in Appendix B. Unless stated otherwise, retrieval evaluation is performed using Top-K pooling, with default values of $k = 40$ for queries and $k = 400$ for documents. For our multilingual models (§ 6), we additionally rely on model averaging Wortsman et al. (2022) from several training runs, which boosts generalization performance (Lee et al., 2025b; Zhang et al., 2025).

We are mainly interested in ***comparing SPLARE to current state-of-the-art LSR methods, which are all vocabulary-based***. To this end, we perform controlled comparisons with a SPLADE model built on the same Llama-3.1-8B backbone—following the methodology of Doshi et al. (2024); Zeng et al. (2025)—and trained under identical settings. We refer to this baseline as SPLADE-Llama.

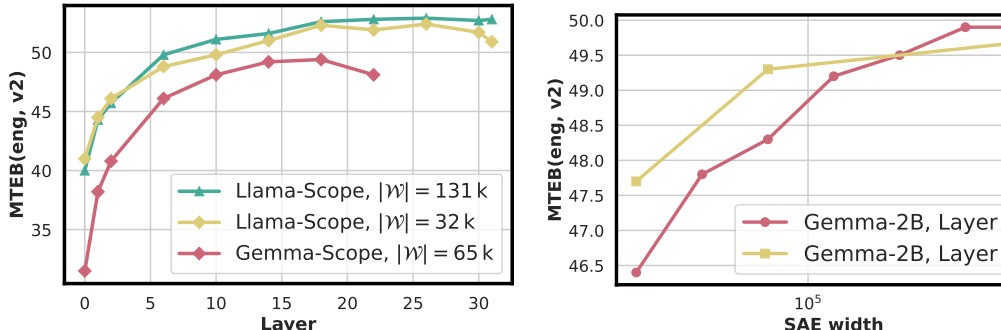

Figure 2: *(Left)* Performance across layers on Llama Scope (Llama-3.1-8B) and Gemma Scope (Gemma-2-2B). *(Right)* Performance with increasing SAE width on Gemma-2. Evaluation done with Top-K $= (40, 400)$.

## 5 ANALYSIS AND DESIGN CHOICES FOR SPLARE MODELS

We first conduct a series of ablation studies in a controlled, English-only setting. At this stage, our primary objective is to compare SPLARE's latent representations with traditional vocabulary-based approaches (i.e., our SPLADE-Llama baseline). Specifically, we aim to address the following research questions: *(i)* At which transformer layer depth do we obtain the most effective sparse latent representations for retrieval? *(ii)* How does the width of the SAE affect retrieval performance? *(iii)* What are the efficiency–effectiveness trade-offs introduced by the latent vocabulary? *(iv)* Do the sparse latent features learned by the SAE yield improvements over equivalent SPLADE models?

**Performance and Layer Depth.** We train SPLARE models at varying depths on Llama-3.1-8B, using SAEs from Llama Scope with two widths $|\mathcal{W}| \in \{32k, 131k\}$, and on Gemma-2-2B, using Gemma Scope with width $|\mathcal{W}| = 65k$, and report the average MTEB (English, v2) performance in Figure 2 *(Left)*. Interestingly, the highest performance is consistently achieved at about two-thirds of the model depth, i.e., around layer 20 (out of 32) for Llama Scope and 16 (out of 26) for Gemma Scope. These findings are consistent with prior work suggesting that intermediate transformer layers often yield richer representations for retrieval tasks (Skean et al., 2025; Zhuang et al., 2025; Wang et al., 2025). A further advantage of using intermediate layers is the reduction in retriever size and, consequently, inference latency—an improvement over SPLADE models, which require processing through all layers of the LLM (see Appendix F).

While the best performance often appears at relatively deep layers, this choice is not critical: performance at earlier layers remains strong, and the selection ultimately reflects an effectiveness-efficiency trade-off. A practical rule of thumb is to select a layer around two-thirds of the model depth to maximize effectiveness. For the remainder of the paper, our main SPLARE models are trained at layer 26 of Llama-3.1-8B, yielding a 7B-parameter model (including the SAE parameters), but we also train a strong 2B model (layer 6) in § 6[3].

**How does the Width of the SAE Affect Retrieval Performance?** Unlike SPLADE models, the dimensionality of SPLARE's feature space—determined by the SAE width $|\mathcal{W}|$—is not constrained by the LLM's vocabulary size. To study the effect of SAE width on retrieval effectiveness, we train multiple SPLARE models using Gemma Scope, which offers a broader range of SAE configurations. Especially, we consider SAEs at layers 12 and 19 of Gemma-2-2B with widths $|\mathcal{W}| \in \{2^{14} \approx 16k, 2^{15}, \ldots, 2^{20} \approx 1M\}$. We report the resulting average MTEB (English, v2) performance in Figure 2 *(Right)*. Our results show a roughly log-linear relationship between SAE width and retrieval effectiveness, providing a scaling mechanism for improved performance—something not possible with SPLADE's fixed vocabulary size. Prior work has shown that SAEs can scale to widths as large as $14M$ on very large LLMs Templeton et al. (2024), though such models remain proprietary. Llama Scope, while limited to $|\mathcal{W}| \in \{32k, 131k\}$, exhibits the same scaling effect consistently

---

[3]Unless otherwise specified, SPLARE refers to SPLARE-7B.

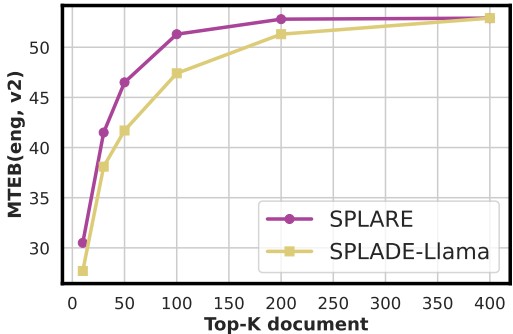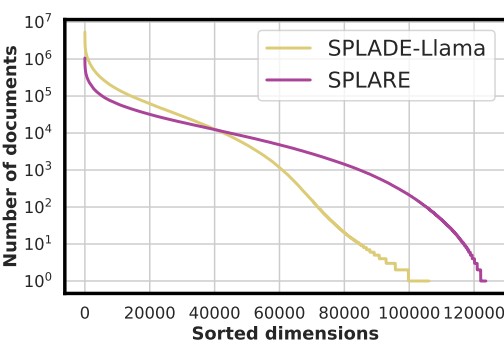

Figure 3: *(Left)* Impact of pruning documents with Top-K (with $k = 40$ for queries). *(Right)* MS MARCO index distribution for SPLARE and SPLADE ($8.8M$ documents).

across layers (Figure 2, *(Left)*). These experiments also highlight that the approach is transferable across different backbone architectures. Despite the availability of much wider SAEs in Gemma Scope, we observe that Llama Scope models achieve superior overall performance. Consequently, we report results using this model (with $|\mathcal{W}| = 131k$) for all subsequent experiments.

**Effectiveness–Efficiency Trade-Off.** Sparse retrieval methods achieve efficiency through the use of dedicated inverted index structures and exact Zobel & Moffat (2006) or approximate Bruch et al. (2024a) query processing algorithms. In all cases, obtaining highly sparse representations is critical for achieving low-latency retrieval. While SPLADE has been successfully adapted to LLM backbones, efficiency considerations have generally been overlooked. As discussed in § 3.2, LSR models can easily become "dense" in practical scenarios, which undermines their efficiency.

We study the relationship between SPLARE performance and sparsity by capping, at inference time, the number of activated features for document vectors using Top-K pooling. Results are shown in Figure 3 *(Left)*. SPLARE exhibits substantially greater robustness to document pruning: when indexing only Top-K $= 100$ document features, its performance drops by merely $2\%$, compared to over $6\%$ for SPLADE. This difference can be partially attributed to SPLARE's more compact and structured latent feature space as well as the fact that SPLADE models based on LLMs are inherently harder to sparsify. As we show in Appendix E, this difference translates into lower query latency at a given accuracy level, when evaluated using Seismic (Bruch et al., 2024a;b; 2025). For reference, performing retrieval with SPLARE (Top-K $= (40, 400)$) on MS MARCO ($8.8M$ documents) requires about $5ms$ per query only—without accounting model inference. Figure 3 *(Right)* further illustrates the distributions of activated features after training. Notably, SPLARE utilizes a much larger portion of the available feature space, activating nearly all dimensions, in contrast to SPLADE, which relies on fewer than $100k$ dimensions (out of $128k$). Moreover, SPLARE exhibits a more balanced activation distribution across features. By comparison, SPLADE tends to over-activate a small subset of dimensions (Mackenzie et al., 2023; Lei et al., 2025).

**Comparison of Lexical and Latent Features.** Finally, we compare the performance of SPLARE with existing top LSR methods trained on the English MS MARCO dataset. In particular, Lion-SP-8B Zeng et al. (2025) represents the most effective contemporary SPLADE adaptation for LLM-based retrieval. We show the results on Table 1 for various splits of MTEB (English Models). First, notice that SPLADE-Llama (our baseline) already significantly outperforms Lion-SP-8B (e.g, +4.4 on MTEB English). We further observe that SPLARE consistently outperforms competing methods on both multilingual and several out-of-domain evaluation sets. In particular, it achieves an improvement of roughly two points on the multilingual split and shows superior performance on the Law and Medical retrieval benchmarks—though its advantage diminishes on the Code and Chemical splits. The observed multilingual generalization from English-only training is unsurprising, given the language-agnostic nature of SAE features. Meanwhile, the performance drop on the Code tasks is likely due to the highly domain-specific nature of code retrieval, which does not align well with the features learned by the SAE (a trend that is further supported by our observations in § 6). To illustrate this behavior, we provide in Appendix G (Figures 15-17) several examples where SPLARE underperforms compared to SPLADE on MTEB Code. In these cases, the top activated

Table 1: Average performance on various MTEB splits. English models are trained on MS MARCO only (§ 5). Multilingual models are trained on a large-scale multilingual training set (§ 6). Evaluation done with Top-K $= (40, 400)$.

| | English | Multilingual | Code | Medical | Law | ChemTEB |
|---|---|---|---|---|---|---|
| **English Models** | | | | | | |
| SPLADE-v3 (Lassance et al., 2024) | 50.7 | 38.1 | 44.5 | 44.2 | 40.4 | 75.6 |
| Lion-SP-8B (Zeng et al., 2025) | 48.5 | 50.0 | 53.3 | 54.4 | 48.5 | 71.1 |
| SPLADE-Llama | **52.9** | 54.3 | **57.3** | 61.0 | 49.0 | **75.9** |
| SPLARE | **52.9** | **56.3** | 55.1 | **62.9** | **51.2** | 70.0 |
| **Multilingual Models** | | | | | | |
| SPLADE-Llama | 58.9 | 61.7 | **64.3** | 67.6 | 60.7 | 77.4 |
| SPLARE | **59.3** | **62.3** | 63.0 | **67.7** | **60.8** | **78.1** |

Table 2: Multilingual comparison of SPLARE and SPLADE (Top-K $= (40, 400)$).

| | indic | sca | deu | fra | kor | XTREME-UP | MIRACL |
|---|---|---|---|---|---|---|---|
| SPLADE-Llama | 91.9 | 70.4 | **57.3** | **65.6** | 74.8 | 56.2 | 69.9 |
| SPLARE | **92.3** | **70.8** | 57.1 | 64.8 | **76.0** | **58.6** | **71.7** |

features appear overly generic rather than specialized to code semantics. This suggests that for highly domain-specific scenarios such as code retrieval, dedicated SAEs trained on code-focused corpora may be more appropriate. We leave this direction for future work.

# 6 MULTILINGUAL MODELS

In § 5, we showed how the latent feature space of the SAE offers some advantages for LSR models—when compared to the vocabulary space—*in a controlled English-based setting*. In § 6.1, we further extend those findings for multilingual models, by training models on a large-scale multilingual dataset and broadening the evaluation to cover a more diverse set of benchmarks, as detailed in § 3.2. In § 6.2, we compare SPLARE to concurrent models on (M)MTEB and XTREME-UP.

## 6.1 COMPARING LATENT MODELS TO LEXICON-BASED APPROACHES

Table 1 (Multilingual models) reports the average performance of multilingual SPLARE and SPLADE across the various MTEB splits, with complete results available in Appendix D. Overall, SPLARE consistently outperforms its vocabulary-based counterpart, except on the Code split—an outcome aligned with the discussion in § 5. Notably, the performance gap is larger on the Multilingual split of MTEB (+0.6 average nDCG@10). Examining the *multilingual-only* datasets within this split confirms that SPLARE systematically surpasses SPLADE (Table 8, +0.9 points). This trend is confirmed by Table 2, which underscores the advantage of latent-based LSR models in multilingual scenarios (e.g., +2.4 on XTREME-UP and +1.8 on MIRACL). Comprehensive results for MIRACL and XTREME-UP, together with comparisons to concurrent methods, are presented in Appendix D and Table 3, respectively. SPLARE achieves particularly strong performance on the hidden test sets of MIRACL (Table 11, de and *yo*) as well as on the low-resource languages of XTREME-UP.

## 6.2 COMPARING TO TOP MODELS

Finally, we compare SPLARE to top models from the MTEB leaderboard in Table 3. SPLARE reaches an average score of 62.3 (for the pooled version), *making it among the top 10 models on MTEB(Multilingual, v2) retrieval and the top-1 LSR model. Notably, these results are achieved without relying on private or synthetic data and without any pre-finetuning*. This is also particularly interesting, as open models like gte-Qwen2-7B instruct or NV-Embed-v2 rely on 3584-$d$ (resp. 4096-$d$) dense vectors to encode queries and documents, while SPLARE only needs 40 features (resp. 400) to encode queries (resp. documents) in its high-dimensional feature space to reach high

Table 3: Average MTEB retrieval performance of SPLARE against top models. Multilingual (resp. Eng) refers to MTEB(Multilingual, v2) (resp. MTEB(eng, v2)). At the time of writing (February 26, 2026), SPLARE ranks in the top-10 models on MTEB(Multilingual, v2) retrieval. For XTREME-UP, we report results from (Lee et al., 2025b). Unless specified, evaluation for SPLARE is done with Top-K $= (40, 400)$—corresponding to our default model SPLARE or SPLARE-2B.

| | English | Multilingual | XTREME-UP |
|---|---|---|---|
| **Top Open Source models** | | | |
| e5-mistral-7b-instruct (Wang et al., 2024a) | 57.6 | 55.8 | - |
| NV-Embed-v2 (Lee et al., 2025a) | 62.8 | 56.7 | - |
| multilingual-e5-large-instruct (Wang et al., 2024b) | 53.5 | 57.1 | 18.7 |
| GritLM-7B (Muennighoff et al., 2024) | 55.0 | 58.3 | - |
| SFR-Embedding-Mistral (Meng et al., 2024) | 59.3 | 59.4 | - |
| Linq-Embed-Mistral (Kim et al., 2024) | 60.1 | 58.7 | 24.6 |
| gte-Qwen2-7B-instruct (Li et al., 2023b) | 58.1 | 60.1 | 17.4 |
| voyage-3-large (AI, 2025) | 53.5 | 66.1 | 39.2 |
| jina-embeddings-v4 (Günther et al., 2025) | 56.2 | 66.4 | - |
| inf-retriever-v1 (Yang et al., 2025) | 64.1 | 66.5 | - |
| Qwen-3-Embedding-8B (Zhang et al., 2025) | 69.4 | 70.9 | - |
| **Commercial models** | | | |
| Cohere-embed-multilingual-v3.0 (Cohere, 2023) | 55.7 | 59.2 | - |
| text-embedding-3-large (OpenAI, 2024) | 58.0 | 59.3 | 18.8 |
| gemini-embedding-001 (Lee et al., 2025b) | 64.4 | 67.7 | 64.3 |
| SPLARE | 59.3 | 62.3 | 58.6 |
| SPLARE - no-pooling | 61.4 | 63.8 | 61.4 |
| SPLARE - Top-K $= (20, 200)$ | 55.9 | 59.9 | 53.8 |
| SPLARE - Top-K $= (10, 100)$ | 50.1 | 56.0 | 46.5 |
| SPLARE-2B | 55.9 | 59.1 | 41.6 |

effectiveness (Top-K at inference time). We also observe an average gain of $+1.5$ points for the non-pooled version, albeit at the cost of higher retrieval complexity. On the other hand, extremely sparse models (Top-K $= (10, 100)$) still offer competitive performance. Note that in practical retrieval scenarios, dense embeddings often require dimensionality-reduction techniques Kusupati et al. (2022) and/or approximate nearest-neighbor search algorithms Johnson et al. (2019)—whose performance degradation is rarely reported on standard benchmarks. In contrast, sparse retrieval methods natively support efficient exact search without incurring such compromises. Finally, we also report results for a SPLARE model trained at layer 6 (SPLARE-2B). Although its performance is somewhat lower than that of the full SPLARE model (7B parameters), it remains strong—particularly on the XTREME-UP dataset. Importantly, this model is substantially more efficient and therefore offers a different, and often attractive, point on the effectiveness–efficiency trade-off curve.

## 6.3 INTERPRETABILITY: MECHANISTIC INTERPRETATION OF SPLARE

Finally, we provide interpretability insights for SPLARE. We leverage Neuronpedia (Lin, 2023) to obtain explanations for individual SAE features—which, as a reminder, remain frozen during fine-tuning—and list the top features contributing to a document's relevance with respect to a given query. For SPLADE, by contrast, we report the tokens with the highest relevance contributions. Figure 4 illustrates a cross-lingual example from XTREME-UP from Tamil to English. The features activated by SPLARE align well with meaningful concepts present in both the query and document. They correspond to coherent, language-agnostic concepts which combine into a comprehensive description of the data point. In contrast, SPLADE exhibits a higher degree of redundancy (e.g., separate activations for "Indian" and "indian") and predominantly relies on Latin-script tokens—effectively defaulting to English subword representations—which provide less informative signals in this setting. Further examples are given in Appendix G.

> **Figure 4: Retrieval example from XTREME-UP: `Tamil → English`**
>
> **`Query`**: அங்கிலேயர்கள் ஆட்சியில் சராசரியாக எத்தனை இந்தியர்கள் இறந்தனர்
>
> **`Translation`**: On average, how many Indians died under British rule?
>
> **`Positive document`**: Indian Army during World War II: The British Indian Army fought in Ethiopia against the Italian Army, in Egypt, Libya, Tunisia and Algeria against both the Italian and German Army, and, after the Italian surrender, against the German Army in Italy. [...]
>
> | **SPLARE | top features** (doc rank = 4) | | **SPLADE | top tokens** (doc rank = 23) | |
> | :--- | ---: | :--- | ---: |
> | Explanation (from Neuronpedia) Lin (2023) | % | Token | % |
> | elements related to historical or cultural contexts | 10.5 | ␣Indian | 12.6 |
> | mentions of India and its relation to various contexts | 8.7 | ␣Indians | 11.2 |
> | descriptions that contrast traditional experiences with unique local accommodation | 7.5 | ␣casualties | 9.0 |
> | mentions of colonial powers, specifically Britain and France | 6.6 | ␣India | 8.5 |
> | references to military casualties and losses | 6.5 | ␣indian | 7.6 |
> | quantitative statistics and casualties related to wars and conflicts | 5.9 | ␣British | 7.5 |
> | information related to economic data and connectivity issues in India | 5.2 | ␣deaths | 6.8 |
> | references to protests and civil rights movements | 5.0 | ␣fatalities | 5.3 |
> | references to historical events and political movements | 4.9 | ␣india | 4.7 |
> | references to corporate structure and business dynamics | 4.4 | ␣Raj | 4.5 |

## 7 RELATED WORKS

**LLMs and Retrieval.** Dense embedding models derived from LLMs have demonstrated substantial gains over traditional BERT-style encoders (Lee et al., 2025b; Zhang et al., 2025). Recent approaches such as LLM2Vec BehnamGhader et al. (2024) or GritLM Muennighoff et al. (2024) highlight how LLMs can be effectively adapted into powerful text encoders by incorporating bidirectional attention. Beyond providing stronger backbone architectures, LLMs have also significantly advanced retrieval model training, enabling the generation of high-quality synthetic data and improved filtering of training samples (Wang et al., 2024a; Lee et al., 2025a;b; Zhang et al., 2025; Dai et al., 2023). Nonetheless, despite the impressive progress of dense embeddings, controlled evaluations have shown that they can still be outperformed by alternative architectures such as multi-vector models or sparse retrievers (Zeng et al., 2025; Faysse et al., 2025; Chen et al., 2024a).

**Sparse Autoencoders and Retrieval.** Sparse autoencoders have primarily been employed in Information Retrieval (IR) to approximate dense representations for efficient nearest-neighbor search. Given a dense embedding model, these approaches learn to map query and document vectors into sparse latent representations that preserve the structure of the original embedding space (Lassance et al., 2021; Borges et al., 2023; Park et al., 2025; Kang et al., 2025; Wen et al., 2025). SAEs have also been used to interpret dense embeddings in both IR O'Neill et al. (2024) and Recommender Systems (Kasalický et al., 2025; Klenitskiy et al., 2025). Most closely related to our work is Park et al. (2025), which shows that SAE-derived features can serve as effective indexing units. However, all prior studies train SAEs on top of an *already-trained dense retriever*. In contrast, our approach leverages pre-trained SAEs on the base LLM and fine-tunes an LSR model directly in a SPLADE-like fashion, allowing for tighter integration of relevance and sparsity.

## 8 CONCLUSION

In this work, we investigated two complementary research directions: Sparse autoencoders and Learned Sparse Retrieval models. We demonstrated that SAEs provide a natural foundation for LSR by yielding semantically rich and multilingual latent features that overcome the vocabulary dependence of traditional LSR approaches. Our experiments show that SAE-based LSR models consistently outperform vocabulary-based counterparts, particularly in multilingual and out-of-domain scenarios. Finally, we introduced SPLARE, a competitive 7B-parameter multilingual model capable of producing generalizable sparse latent embeddings, thereby paving the way for more robust, versatile, and cross-lingual retrieval across diverse domains and modalities.

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

## A  EXPERIMENTAL SETTING

We detail below the training sets used for the English and Multilingual settings.

**English Setting.**  For our ablation study, we restrict training to the MS MARCO dataset, given the computational cost associated with training 7B-parameter models. Our experimental setup closely follows that of SPLADE-v3 (Lassance et al., 2024). For each training query, we mine hard negatives using a SPLADE model and derive distillation targets from reranking scores produced by an open-source DeBERTa-v3 reranker (Déjean et al., 2024). This controlled setting is designed to enable a direct and fair comparison between SPLARE and its vocabulary-based counterpart, SPLADE-Llama.

**Multilingual Setting.**  In this more compute-intensive setting, we use the same training set employed for the bge-multilingual-gemma2 model (Li et al., 2025)[4]. This corpus includes several English-centric public datasets (e.g., MS MARCO Bajaj et al. (2018), NQ Kwiatkowski et al. (2019), and HotPotQA Yang et al. (2018)), a large collection of Chinese retrieval datasets, and two multilingual benchmarks: MIRACL Zhang et al. (2023) and Mr.TyDi (Zhang et al., 2021). Since we rely on distillation for training, we only keep samples from this dataset which were annotated using the BGE multilingual reranker (Chen et al., 2024b; Li et al., 2023a)[5]. After filtering, the final training set comprises approximately $1.3M$ queries with hard negatives. Notably, some of these datasets correspond to training splits of several MTEB benchmark tasks. While this may constrain the strict evaluation of generalization, this practice has become standard in prior work on general-purpose embedding models (Lee et al., 2025a; Wang et al., 2024a; BehnamGhader et al., 2024).

## B  HYPER-PARAMETERS

Table 4 gives the hyper-parameters used to train and evaluate SPLARE models and other baselines. Note that the temperature parameter $\tau$ is critical and needs to be adapted to each SAE suite. For instance, the optimal $\tau$ is different between Llama Scope or Gemma Scope. This depends on the scale of the logits and the initial sparsity of the SAE. For ill-suited $\tau$, it can happen that models actually diverge—for instance, collapse of the $\ell_0$. To determine the optimal temperature, we ran a grid search over the values $\{1, 10, 20, 40, 50, 80, 100\}$, and used NanoBEIR's nDCG@10 as an evaluation criterion for all models[6].

**SAE Choice.**  Gemma Scope contains multiple SAEs for the same layer and width, but with different $\ell_0$. In practice, we observed that the initial SAE's $\ell_0$ had no critical effect on final performance—most likely because we fine-tune the backbone LLM. We use SAEs with $\ell_0$ closest to 100 throughout the paper. Additionally, Llama and Gemma Scope contain residual SAEs as well as MLP and attention stream SAEs—we only use residual SAEs in this study.

## C  ENGLISH-ONLY SPLARE FULL RESULTS

We evaluate models from Section 5 (trained on English data only) on several benchmarks, and provide results in Table 1. Table 5 additionally reports evaluation results comparing SPLARE with SPLADE-Llama. We report MRR@10 on MS MARCO Bajaj et al. (2018) and nDCG@10 on TREC DL datasets Craswell et al. (2021) as well as on all BEIR datasets (Thakur et al., 2021).

## D  FULL RESULTS

Tables 6—10 provide the full results of several MTEB datasets: English, Multilingual, and various domains and languages.

---

[4] `hanhainebula/bge-multilingual-gemma2-data`

[5] `BAAI/bge-reranker-v2-m3` reranker

[6] `https://huggingface.co/collections/zeta-alpha-ai/nanobeir-66e1a0af21dfd93e620cd9f6`

Table 4: Hyper-parameters.

| Component | Value |
|---|---|
| LoRA rank $r$ | 64 |
| Max training sequence length (english models) | 128 |
| Max training sequence length (multilingual models) | 512 |
| Epochs | 1 |
| Batch size (w/ gradient accumulation) | 128 |
| Learning rate | $5 \times 10^{-5}$ |
| Warmup ratio | 0.01 |
| Nb negatives per query | 8 |
| $\lambda_d$ | 0.0001 |
| $\lambda_q$ | 0.0001 |
| $\tau$ SPLARE - Llama Scope | 80 |
| $\tau$ SPLARE - Gemma Scope | 50 |
| $\tau$ SPLADE-Llama | 10 |
| Evaluation max context size | 1024 |
| Adam $\beta$s | 0.9, 0.999 |

| Dataset | SPLARE | SPLADE-Llama |
|---|---|---|
| Arguana | 16.0 | 16.2 |
| Climate-FEVER | 18.3 | 18.0 |
| DBPedia | 44.3 | 44.8 |
| FEVER | 76.0 | 75.8 |
| FiQA-2018 | 42.4 | 42.3 |
| HotpotQA | 66.8 | 67.6 |
| NFCorpus | 37.3 | 36.4 |
| NQ | 61.6 | 61.2 |
| Quora | 87.3 | 87.9 |
| SCIDOCS | 17.5 | 17.3 |
| SciFact | 72.5 | 72.9 |
| TREC-COVID | 84.7 | 82.4 |
| Touché-2020 | 27.2 | 26.9 |
| Average | 50.2 | 50.0 |
| MS MARCO (MRR@10) | 40.8 | 40.0 |
| TREC DL '19 | 77.4 | 76.3 |
| TREC DL '20 | 77.3 | 75.9 |

Table 5: Full results (nDCG@10 unless specified) on BEIR, MS MARCO and TREC DL for English-based SPLARE and SPLADE-Llama models. Evaluation done with Top-K $= (40, 400)$.

Table 11 compares the SPLARE results on the MIRACL dataset with top multilingual dense retrievers—baseline results are taken from Chen et al. (2024b). On this benchmark, SPLARE obtains an average score of 71.7, +0.2 points above M3-embeddings (hybrid: dense + sparse) Chen et al. (2024a). Notably, SPLARE is state-of-the-art in English, Finnish, Hindi, Japanese, Russian, Swahili, German and Yoruba, once again indicating its ability to generalize to diverse languages. Note in particular that German and Yoruba are the "secret" languages of MIRACL which were released later *without associated training data*.

## E  LATENCY MEASURES

We provide per-query retrieval latency as measured on MS MARCO (retrieval from a collection of $8.8M$ documents) for SPLARE and SPLADE-Llama in Figure 5. To measure this, we first index the collection using Seismic Bruch et al. (2024c), and then perform single-threaded retrieval on the

| Task Name | SPLARE | SPLADE-Llama |
|---|---|---|
| ArguAna | 67.4 | 65.9 |
| CQADupstackGamingRetrieval | 59.6 | 59.3 |
| CQADupstackUnixRetrieval | 43.7 | 44.9 |
| ClimateFEVERHardNegatives | 33.4 | 35.9 |
| FEVERHardNegatives | 90.6 | 91.1 |
| FiQA2018 | 57.8 | 57.2 |
| HotpotQAHardNegatives | 76.0 | 73.2 |
| SCIDOCS | 20.8 | 20.3 |
| TRECCOVID | 83.4 | 82.6 |
| Touche2020Retrieval.v3 | 60.8 | 58.4 |
| Average | 59.3 | 58.9 |

Table 6: Full results of SPLARE and SPLADE-Llama on MTEB(Eng, v2). Evaluation done with Top-K $= (40, 400)$.

| Task Name | SPLARE | SPLADE-Llama |
|---|---|---|
| AILAStatutes | 38.9 | 36.3 |
| ArguAna | 67.4 | 65.9 |
| BelebeleRetrieval | 83.9 | 83.8 |
| CovidRetrieval | 83.3 | 81.5 |
| HagridRetrieval | 98.9 | 98.8 |
| LEMBPasskeyRetrieval | 48.2 | 48.2 |
| LegalBenchCorporateLobbying | 95.1 | 94.9 |
| MIRACLRetrievalHardNegatives | 72.4 | 70.5 |
| MLQARetrieval | 83.8 | 81.5 |
| SCIDOCS | 20.8 | 20.3 |
| SpartQA | 5.3 | 5.2 |
| StackOverflowQA | 88.4 | 90.0 |
| StatcanDialogueDatasetRetrieval | 30.8 | 30.5 |
| TRECCOVID | 83.4 | 82.6 |
| TempReasonL1 | 2.4 | 3.8 |
| TwitterHjerneRetrieval | 73.2 | 73.4 |
| WikipediaRetrievalMultilingual | 91.7 | 90.6 |
| WinoGrande | 53.1 | 52.8 |
| Average | 62.3 | 61.7 |

Table 7: Full results of SPLARE and SPLADE-Llama on MTEB(Multilingual, v2). Evaluation done with Top-K $= (40, 400)$.

saved index. Building a highly optimized sparse retrieval setup is difficult in general; here we use the Seismic library with default hyperparameters—given in Table 14.

In this simple setup, retrieval takes around 5*ms* per query with maximal accuracy for SPLARE. In low-latency regime ($< 4ms$), SPLARE can be used with higher accuracy compared to SPLADE.

## F   SPLADE LAYER ABLATION

In § 5, we showed that SPLARE models are typically more effective at intermediate layers, yielding a latency advantage over SPLADE. In principle, however, SPLADE models can also be trained using intermediate representations by simply applying the LM head to these layers. Table 15 reports results obtained with this training procedure.

| Task Name | SPLARE | SPLADE-Llama |
|-----------|--------|--------------|
| BelebeleRetrieval | 83.9 | 83.8 |
| MIRACLRetrievalHardNegatives | 72.4 | 70.5 |
| MLQARetrieval | 83.8 | 81.5 |
| StatcanDialogueDatasetRetrieval | 30.8 | 30.5 |
| TwitterHjerneRetrieval | 73.2 | 73.4 |
| WikipediaRetrievalMultilingual | 91.7 | 90.6 |
| Average | 72.6 | 71.7 |

Table 8: Full results of SPLARE and SPLADE-Llama on the *multilingual-only* datasets of MTEB(Multilingual, v2). Evaluation done with Top-K $= (40, 400)$.

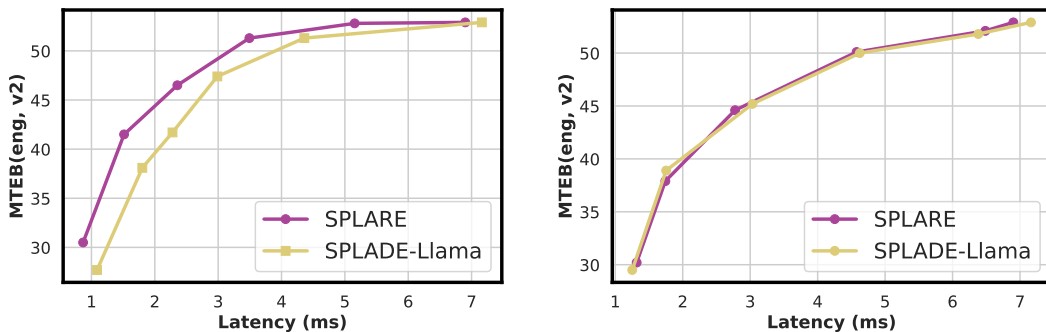

Figure 5: Retrieval Latency (*ms*) when pooling documents *(Left)* or query *(Right)* representations with Top-K. In low-latency settings, SPLARE enables higher accuracy for a given level of latency.

---

**Figure 6: Retrieval example from `BEIR/Scifact`**

**Query**: Flexible molecules experience greater steric hindrance in the tumor microenviroment than rigid molecules.

**Positive document**: A solid tumor is an organ composed of cancer and host cells embedded in an extracellular matrix and nourished by blood vessels. A prerequisite to understanding tumor pathophysiology is the ability to distinguish and monitor each component in dynamic studies. Standard fluorophores hamper simultaneous intravital imaging of these components. Here, we used multiphoton microscopy techniques and transge [...]

| SPLARE \| top features (doc rank = 3) Explanation (from Neuronpedia) Lin (2023) | % | SPLADE \| top tokens (doc rank = 6) Token | % |
|---|---|---|---|
| references to tumors and their related biological processes | 8.6 | ␣tumor | 12.7 |
| terms related to drug delivery and cellular mechanisms | 7.5 | ␣tumors | 8.1 |
| terms related to cancer research and metastasis | 5.5 | ␣cancer | 7.3 |
| medical conditions and diseases, particularly types of cancer and their characteristics | 4.8 | ␣tum | 6.3 |
| concepts related to flexibility in various contexts | 4.8 | ␣nanoparticles | 5.3 |
| terms related to cellular processes and immune system functions | 4.2 | ␣Cancer | 4.9 |
| references to experimental methods and cell-related terminology | 4.0 | ␣nanop | 4.4 |
| terms related to microscopy and micro-level scientific analysis | 4.0 | ␣nan | 3.0 |
| variations of the word "tumble" or its related forms | 3.6 | ␣solid | 2.6 |
| terms related to cancer and tumors | 3.5 | ␣malignant | 2.6 |

---

## G    RETRIEVAL EXAMPLES

We provide in Figures 6—14 multiple examples of scores and explanations obtained for positive documents for some queries on English, Multilingual and multi-domain datasets. We also provide examples on the code domain (Figures 15—17), which highlight some of the limitations of SPLARE on specific domains which might require dedicated SAEs. Notably, in Figure 14 which shows a Tamil example, *the document and query representations coincide for only 6 tokens*, further highlighting SPLADE multilingual limitations. Note that the explanations we used, taken from Neuronpedia, are mostly annotated by LLMs provided with examples of context with features activations. As such, these explanations may remain inaccurate or incomplete.

| Task Name | SPLARE | SPLADE-Llama |
|---|---|---|
| **Code** | | |
| AppsRetrieval | 28.2 | 30.7 |
| COIRCodeSearchNetRetrieval | 62.1 | 70.0 |
| CodeEditSearchRetrieval | 74.2 | 75.0 |
| CodeFeedbackMT | 56.3 | 55.5 |
| CodeFeedbackST | 78.0 | 78.8 |
| CodeSearchNetCCRetrieval | 61.3 | 64.4 |
| CodeSearchNetRetrieval | 85.3 | 86.8 |
| CodeTransOceanContest | 86.7 | 88.5 |
| CodeTransOceanDL | 36.3 | 33.4 |
| CosQA | 31.1 | 30.7 |
| StackOverflowQA | 88.4 | 90.0 |
| SyntheticText2SQL | 68.2 | 67.5 |
| Average | 63.0 | 64.3 |
| **Medical** | | |
| CUREv1 | 61.2 | 57.2 |
| CmedqaRetrieval | 30.2 | 31.5 |
| MedicalQARetrieval | 74.7 | 75.7 |
| NFCorpus | 38.8 | 38.5 |
| PublicHealthQA | 86.7 | 86.2 |
| SciFact | 77.5 | 77.6 |
| SciFact-PL | 73.6 | 74.0 |
| TRECCOVID | 83.4 | 82.6 |
| TRECCOVID-PL | 82.8 | 84.8 |
| Average | 67.7 | 67.6 |
| **Law** | | |
| AILACasedocs | 39.0 | 41.1 |
| AILAStatutes | 38.9 | 36.3 |
| GerDaLIRSmall | 34.7 | 35.8 |
| LeCaRDv2 | 62.3 | 61.3 |
| LegalBenchConsumerContractsQA | 87.4 | 87.9 |
| LegalBenchCorporateLobbying | 95.1 | 94.9 |
| LegalQuAD | 62.2 | 60.8 |
| LegalSummarization | 66.9 | 67.1 |
| Average | 60.8 | 60.7 |
| **ChemTEB** | | |
| ChemHotpotQARetrieval | 86.2 | 85.3 |
| ChemNQRetrieval | 70.0 | 69.6 |
| Average | 78.1 | 77.4 |

Table 9: Full results of SPLARE and SPLADE-Llama on MTEB domain specific datasets. Evaluation done with Top-K $= (40, 400)$.

| Task Name | SPLARE | SPLADE-Llama |
|---|---|---|
| **MTEB(Indic, v1)** | | |
| BelebeleRetrieval | 87.1 | 87.0 |
| XQuADRetrieval | 97.5 | 96.9 |
| Average | 92.3 | 91.9 |
| **MTEB(Scandinavian, v1)** | | |
| DanFeverRetrieval | 42.8 | 41.7 |
| NorQuadRetrieval | 23.6 | 26.5 |
| SNLRetrieval | 98.0 | 98.0 |
| SweFaqRetrieval | 79.4 | 77.0 |
| SwednRetrieval | 84.1 | 82.5 |
| TV2Nordretrieval | 94.0 | 94.0 |
| TwitterHjerneRetrieval | 73.2 | 73.4 |
| Average | 70.8 | 70.4 |
| **MTEB(deu, v1)** | | |
| GerDaLIR | 18.9 | 19.3 |
| GermanDPR | 87.5 | 86.7 |
| GermanQuAD-Retrieval | 97.2 | 96.8 |
| XMarket | 24.8 | 26.3 |
| Average | 57.1 | 57.3 |
| **MTEB(fra, v1)** | | |
| AlloprofRetrieval | 56.4 | 58.4 |
| BSARDRetrieval | 64.4 | 59.9 |
| MintakaRetrieval | 46.2 | 57.7 |
| SyntecRetrieval | 90.7 | 87.4 |
| XPQARetrieval | 66.4 | 64.6 |
| Average | 64.8 | 65.6 |
| **MTEB(kor, v1)** | | |
| Ko-StrategyQA | 82.7 | 83.1 |
| MIRACLRetrieval | 69.4 | 66.5 |
| Average | 76.0 | 74.8 |

Table 10: Full results of SPLARE and SPLADE-Llama on MTEB language-specific benchmarks. Evaluation done with Top-K $= (40, 400)$.

| Model | ar | bn | en | es | fa | fi | fr | hi | id | ja | ko | ru | sw | te | th | zh | de[†] | yo[†] | Avg |
|---|---|---|---|---|---|---|---|---|---|---|---|---|---|---|---|---|---|---|---|
| **Baselines (Prior Work)** | | | | | | | | | | | | | | | | | | | |
| BM25 | 39.5 | 48.2 | 26.7 | 7.7 | 28.7 | 45.8 | 11.5 | 35.0 | 29.7 | 31.2 | 37.1 | 25.6 | 35.1 | 38.3 | 49.1 | 17.5 | 12.0 | 56.1 | 31.9 |
| mDPR | 49.9 | 44.3 | 39.4 | 47.8 | 48.0 | 47.2 | 43.5 | 38.3 | 27.2 | 43.9 | 41.9 | 40.7 | 29.9 | 35.6 | 35.8 | 51.2 | 49.0 | 39.6 | 41.8 |
| mContriever | 52.5 | 50.1 | 36.4 | 41.8 | 21.5 | 60.2 | 31.4 | 28.6 | 39.2 | 42.4 | 48.3 | 39.1 | 56.0 | 52.8 | 51.7 | 41.0 | 40.8 | 41.5 | 43.1 |
| mE5$_{large}$ | 76.0 | 75.9 | 52.9 | 52.9 | 59.0 | 77.8 | 54.5 | 62.0 | 52.9 | 70.6 | 66.5 | 67.4 | 74.9 | 84.6 | 80.2 | 56.0 | 56.4 | 78.3 | 66.6 |
| E5$_{mistral-7b}$ | 73.3 | 70.3 | 57.3 | 52.2 | 52.1 | 74.7 | 55.2 | 52.1 | 52.7 | 66.8 | 61.8 | 67.7 | 68.4 | 73.9 | 74.0 | 54.0 | 54.1 | 79.7 | 63.4 |
| Gemini Embedding | 78.3 | 79.0 | 58.7 | 57.0 | 60.9 | 78.0 | 55.6 | 65.4 | 54.3 | 75.1 | 68.9 | 73.4 | 81.0 | 80.5 | 80.8 | 65.7 | 59.8 | 88.8 | 70.1 |
| M3-Emb (Sparse) | 67.1 | 68.9 | 43.8 | 38.6 | 45.1 | 65.4 | 35.3 | 48.2 | 48.9 | 56.1 | 61.5 | 44.5 | 57.9 | 79.1 | 70.9 | 36.1 | 32.5 | 70.0 | 53.9 |
| M3-Emb (All) | 80.2 | 81.5 | 59.6 | 59.7 | 63.4 | 80.4 | 61.2 | 63.3 | 59.0 | 75.2 | 72.1 | 71.7 | 79.6 | 88.1 | 83.7 | 64.9 | 59.8 | 83.5 | 71.5 |
| SPLADE-Llama | 78.0 | 77.5 | 58.8 | 56.0 | 59.8 | 79.9 | 58.5 | 61.7 | 57.1 | 75.3 | 66.1 | 72.8 | 80.1 | 82.4 | 81.0 | 61.8 | 58.2 | 92.5 | 69.9 |
| SPLARE-7B | 79.7 | 79.9 | 60.9 | 58.5 | 62.1 | 81.4 | 60.5 | 65.5 | 57.6 | 75.9 | 69.5 | 74.1 | 81.8 | 83.2 | 83.1 | 64.1 | 62.5 | 90.0 | 71.7 |
| SPLARE-2B | 75.4 | 67.4 | 53.3 | 55.0 | 54.8 | 75.5 | 56.0 | 59.1 | 54.9 | 67.9 | 67.4 | 65.3 | 69.7 | 77.8 | 76.7 | 59.0 | 56.1 | 78.0 | 65.0 |

Table 11: Multilingual retrieval performance on MIRACL dev (nDCG@10). Baseline results are taken from Chen et al. (2024b) and (Lee et al., 2025b). [†] denotes the two hidden test sets of MIRACL. Evaluation for SPLARE and SPLADE-Llama done with Top-K $= (40, 400)$.

| Model | MRR@10 |
|---|---|
| SPLARE | 58.6 |
| SPLARE (Eng Only) | 41.6 |
| SPLADE-Llama | 56.2 |
| SPLADE-Llama (Eng Only) | 30.5 |
| Gemini Embedding | 64.3 |
| Gemini Embedding (Eng Only) | 49.3 |
| Gecko i18n Embedding | 35.0 |
| voyage-3-large | 39.2 |
| Linq-Embed-Mistral | 24.6 |
| multilingual-e5-large-instruct | 18.7 |
| gte-Qwen2-7B-instruct | 17.4 |
| text-embedding-3-large | 18.8 |

Table 12: XTREME-UP dataset (MRR@10) - Average Scores. Baselines taken from (Lee et al., 2025b). Evaluation for SPLARE done with Top-K = $(40, 400)$.

| Model | Avg. | as | bho | brx | gbm | gom | gu | hi | hne | kn | mai | ml | mni | mr | mwr | or | pa | ps | sa | ta | ur |
|---|---|---|---|---|---|---|---|---|---|---|---|---|---|---|---|---|---|---|---|---|---|
| **Eng Only** | | | | | | | | | | | | | | | | | | | | | |
| SPLARE | 42.6 | 44.4 | 49.4 | 10.2 | 48.2 | 39.2 | 40.9 | 57.0 | 49.5 | 46.5 | 52.6 | 47.6 | 20.1 | 50.8 | 49.8 | 26.3 | 44.4 | 37.3 | 43.5 | 45.6 | 48.6 |
| SPLADE-Llama | 30.5 | 31.7 | 46.8 | 10.7 | 45.8 | 30.7 | 24.2 | 54.7 | 44.5 | 19.0 | 48.7 | 21.8 | 20.0 | 45.2 | 48.2 | 7.6 | 23.3 | 2.2 | 38.5 | 27.8 | 19.5 |
| **Multilingual** | | | | | | | | | | | | | | | | | | | | | |
| SPLARE | 58.6 | 63.3 | 63.1 | 14.2 | 60.8 | 58.8 | 65.1 | 67.2 | 63.2 | 64.7 | 65.0 | 68.9 | 27.4 | 64.5 | 63.3 | 54.5 | 66.1 | 52.7 | 63.8 | 63.3 | 61.2 |
| SPLADE-Llama | 56.2 | 61.2 | 58.8 | 14.2 | 59.4 | 56.5 | 62.4 | 64.1 | 62.5 | 62.2 | 64.5 | 63.2 | 29.7 | 60.5 | 59.6 | 53.2 | 62.2 | 50.7 | 62.7 | 59.0 | 57.6 |

Table 13: XTREME-UP dataset (MRR@10) - Full scores. Evaluation done with Top-K = $(40, 400)$.

| Parameter | Value |
|---|---|
| k | 1000 |
| query_cut | 30 |
| heap_factor | 0.5 |
| n_knn | 0 |
| sorted | False |
| num_threads | 1 |

Table 14: Seismic retrieval parameters used to measure latency.

| SPLADE-Llama at intermediate layers | | | | |
|---|---|---|---|---|
| Layer No. | 18 | 22 | 26 | 31 |
| MTEB(Eng, v2) | 0. | 43.6 | 44.5 | 52.9 |

Table 15: Training SPLADE-Llama models at intermediate layers leads to strong deterioration. At layer $< 22$, models collapse during training.

---

**Figure 7: Retrieval example from `BEIR/Scifact`**

`Query`: PPAR-RXRs are inhibited by PPAR ligands.

`Positive document`: Heterodimerization is a common paradigm among eukaryotic transcription factors. The 9-cis retinoic acid receptor (RXR) serves as a common heterodimerization partner for several nuclear receptors, including the thyroid hormone receptor (T3R) and retinoic acid receptor (RAR). This raises the question as to whether these complexes possess dual hormonal responsiveness. We devised a strategy to examine . . .

| SPLARE \| top features (doc rank = 5) Explanation (from Neuronpedia) Lin (2023) | % | SPLADE \| top tokens (doc rank = 18) Token | % |
|---|---|---|---|
| terms related to gene transcription regulation | 6.2 | ␣heter | 6.2 |
| mathematical variables and expressions | 4.3 | ␣RX | 6.0 |
| terms related to dopamine and receptor interactions in the context of medicine and psycho . . . | 4.2 | rx | 5.2 |
| abbreviations or terms related to programming and data structures | 4.2 | RX | 5.0 |
| references to QR codes and VR technologies | 4.2 | ␣receptor | 4.9 |
| information about medications used for treating acne | 3.8 | ␣receptors | 4.3 |
| references to proteins and their biological functions | 3.1 | ␣nuclear | 3.8 |
| terminology related to pharmaceuticals and drug development | 2.8 | ␣Rx | 3.7 |
| terms related to cellular functions and regulatory mechanisms | 2.8 | ␣transcription | 3.6 |
| terms related to medical and biological concepts, particularly hormones and their effects | 2.8 | ␣rx | 3.6 |

---

**Figure 8: Retrieval example from `BEIR/Climate-Fever`**

`Query`: Ocean acidification is the terrifying threat whereby all that man-made CO2 we've been pumping into the atmosphere may react with the sea to form a sort of giant acid bath.

`Positive document`: A greenhouse gas ( abbrev . GHG ) is a gas in an atmosphere that absorbs and emits radiation within the thermal infrared range . This process is the fundamental cause of the greenhouse effect . The primary greenhouse gases in Earth 's atmosphere are water vapor , carbon dioxide , methane , nitrous oxide , and ozone . Without greenhouse gases , the average temperature of Earth 's surface would be a . . .

| SPLARE \| top features (doc rank = 6) Explanation (from Neuronpedia) Lin (2023) | % | SPLADE \| top tokens (doc rank = 20) Token | % |
|---|---|---|---|
| references to climate change and its associated causes | 6.3 | ␣CO | 8.2 |
| statements and discussions regarding climate change-related issues | 5.0 | ␣atmosphere | 7.7 |
| terms related to climate change and its impacts | 4.4 | ␣greenhouse | 7.3 |
| content related to environmental impacts, particularly concerning carbon dioxide and food . . . | 4.1 | ␣climate | 5.7 |
| terms related to carbon emissions and environmental impacts | 4.1 | ␣carbon | 5.6 |
| mentions of carbon dioxide and its related metrics or expressions | 3.9 | ␣dioxide | 4.7 |
| references to carbon dioxide and its implications in various contexts | 3.3 | ␣anthrop | 4.2 |
| references to environmental impact and sustainability | 3.2 | ␣Climate | 3.6 |
| references to human activity and its impact on the environment | 3.0 | ␣atmospheric | 3.6 |
| mentions of sustainability and environmental impact | 3.0 | ␣Carbon | 3.5 |

---

**Figure 9: Retrieval example from `BEIR/Climate-fever`**

`Query`: No state generates as much solar power as California, or has as many people whose jobs depend on it.

`Positive document`: California is the most populous state in the United States and the third most extensive by area . Located on the western ( Pacific Ocean ) coast of the U.S. , California is bordered by the other U.S. states of Oregon , Nevada , and Arizona and shares an international border with the Mexican state of Baja California . The state capital is Sacramento . Los A . . .

| SPLARE \| top features (doc rank = 2) Explanation (from Neuronpedia) Lin (2023) | % | SPLADE \| top tokens (doc rank = 6) Token | % |
|---|---|---|---|
| references to financial or budgetary topics | 7.0 | ␣California | 8.5 |
| references to California | 6.8 | ␣california | 6.3 |
| regional references and mentions of cities or places | 5.7 | ␣CA | 6.1 |
| references to California and its locations or institutions | 4.9 | ␣Calif | 4.9 |
| mentions of political entities and territories | 4.3 | ␣state | 4.7 |
| references to political figures and legislation related to California | 3.9 | ␣Californ | 4.7 |
| references to geographic locations and regions in California, particularly related to agri . . . | 3.8 | California | 4.6 |
| references to governance, laws, and political contexts | 3.3 | ifornia | 3.6 |
| positive descriptions and references to favorable weather conditions | 3.2 | ␣CAL | 3.3 |
| references to California's environmental regulatory bodies and legislation | 3.2 | ␣State | 3.2 |

**Figure 10: Retrieval example from `BEIR/Hotpotqa`**

**Query**: The Death of Cook depicts the death of James Cook at a bay on what coast?
**Positive document**: Kealakekua Bay is located on the Kona coast of the island of Hawaii about 12 mi south of Kailua-Kona.

| SPLARE \| top features (doc rank = 3) Explanation (from Neuronpedia) Lin (2023) | % | SPLADE \| top tokens (doc rank = 17) Token | % |
|---|---|---|---|
| references to "Bay" or similar geographical features | 11.9 | ␣Hawaii | 9.9 |
| references to a specific geographical location or name containing "Bay." | 10.7 | ␣bay | 9.7 |
| geographical features and safe navigation routes | 9.5 | ␣Hawai | 9.1 |
| references to health and community support systems | 9.5 | ␣Bay | 8.2 |
| historical references and significant events | 9.3 | ␣Ke | 7.9 |
| references to coastal regions and their characteristics | 7.6 | Ke | 5.4 |
| references to historical sites and landmarks | 5.4 | Bay | 5.1 |
| references to specific geographical locations and their significance in the context of li . . . | 5.2 | ␣Hawaiian | 5.0 |
| information related to marine and coastal ecosystems | 4.7 | bay | 4.5 |
| references to sailing, ships, and boating experiences | 4.2 | ␣Haw | 4.0 |

**Figure 11: Retrieval example from `MIRACL/Swahili`**

**Query**: Kiongozi wa chama cha Orange Democratic Movement ni nani?
**Positive document**: Orange Democratic Movement Katika uchaguzi wa rais Raila Odinga alitangazwa kuwa ameshindwa na rais Kibaki kwa kura 230,000. Lakini watazamaji wengi waliona kasoro katika hesabu ya kura na ODM ilidai kuwa Odinga ni mshindi halali. ODM ilifaulu vizuri upande wa viti vya bunge la Kenya. Ilipata karibu nusu ya wabunge wote yaani 99 kati ya 120 ikawa kubwa katika bunge baada ya uchaguzi wa Desemba 200 . . .

| SPLARE \| top features (doc rank = 5) Explanation (from Neuronpedia) Lin (2023) | % | SPLADE \| top tokens (doc rank = 8) Token | % |
|---|---|---|---|
| mentions of "Orange" or related terms and concepts | 8.9 | ␣Rail | 9.3 |
| references to events or occurrences in the future | 6.4 | ␣Orange | 8.5 |
| prominent political figures and their involvement in elections | 6.0 | ␣Kenya | 7.1 |
| references to the abbreviation "OD" and variations of it, typically related to a specific . . . | 5.8 | Rail | 6.3 |
| terms associated with political events and discussions | 5.4 | OD | 5.7 |
| references to business strategies and company operations | 5.1 | ␣OD | 5.5 |
| references to DMCA regulations and related legal terms | 4.1 | Orange | 4.8 |
| references to places in Kenya | 4.0 | ␣movement | 4.2 |
| information related to notable historical figures and their relationships | 3.6 | ␣leader | 4.0 |
| references to political candidates and their activities within the Democratic Party | 3.3 | ␣orange | 3.5 |

**Figure 12: Retrieval example from `MIRACL/Bengali`**

**Query (translated from Bengali)**: What is the name of the first band in Bangladesh?
**Positive document (translated from Bengali)**: Obscure (Bangla Band) — Obscure is one of the notable bands in the history of Bangladeshi band music. In the 1980s, Sayed Hasan Tipu took the initiative to establish this band. On March 15, 1985, Tipu founded Obscure in Khulna. During the 1980s, Obscure's first album was released from Sargam Studio. That first self-titled album, "Obscure Volume 1," released in 1986, earned a permanent place in the history of Bangla band music.

| SPLARE \| top features (doc rank = 3) Explanation (from Neuronpedia) Lin (2023) | % | SPLADE \| top tokens (doc rank = 20) Token | % |
|---|---|---|---|
| references to specific individuals and groups within a social or cultural context | 6.2 | ␣Bangladesh | 9.8 |
| references to musical bands or groups | 6.2 | ␣band | 8.5 |
| mentions of bands and musical groups | 5.5 | ␣Bang | 7.9 |
| repeated or emphasized mentions of specific entities or concepts | 4.8 | Bang | 5.8 |
| references to iconic rock bands and their legacy | 4.7 | ␣bang | 5.5 |
| occurrences of the country name "Bangladesh." | 4.7 | ␣Band | 5.3 |
| references to musical bands and collaborations | 4.5 | band | 4.8 |
| elements related to music and musicians | 3.4 | ␣bands | 4.2 |
| descriptors related to music and performance quality | 3.3 | bang | 3.8 |
| proper names and the mention of individuals in the text | 3.2 | -band | 3.3 |

**Figure 13: Retrieval example from `MIRACL/French`**

**Query**: Qui est le mathématicien le plus célèbre au monde?

**Positive document**: Nira Chamberlain En 2017, il intervient dans l'atelier du New Scientist "Le monde mathématique". En 2018, il est nommé "mathématicien le plus intéressant du monde" par le "Big Internet Math Off" organisé par le site "Aperiodical". En 2019, il donne une conférence à la "Maxwell Society" sur "Les mathématiques qui peuvent arrêter une apocalypse de l'IA". Il fait des apparitions dans les médias brita . . .

| SPLARE \| top features (doc rank = 5) Explanation (from Neuronpedia) Lin (2023) | % | SPLADE \| top tokens (doc rank = 11) Token | % |
|---|---|---|---|
| concepts related to mathematics and quantitative analysis | 9.7 | ␣mathematic | 19.7 |
| terms and phrases related to mathematics | 9.0 | ␣Mathematic | 15.3 |
| elements related to academic papers and research acknowledgments | 8.5 | ␣maths | 10.3 |
| references to the concept of "world" in various contexts | 8.3 | ␣math | 10.2 |
| references to mathematical concepts and theorems | 6.8 | ␣Math | 8.1 |
| discussions about artistic individuals or the concept of creativity | 6.6 | ian | 7.9 |
| terms and references related to mathematics and mathematicians | 6.2 | ␣world | 6.6 |
| references to mathematical concepts and terms | 6.1 | ␣monde | 5.9 |
| terms related to academic professionals and researchers across various fields | 5.1 | ␣mathematical | 4.4 |
| references to notable individuals and their contributions or warnings in the field of arti . . . | 5.0 | ematik | 3.9 |

**Figure 14: Retrieval example from XTREME-UP: `Tamil → English`**

**Query**: மனிதனால் சராசரியாக எவ்வளவு வெட்பநிலையை தாங்க முடியும்

**Translation**: On average, how much temperature can a human withstand?

**Positive document**: Cold and heat adaptations in humans The human body always works to remain in homeostasis. One form of homeostasis is thermoregulation. Body temperature varies in every individual, but the average internal temperature is 37.0 ℃ (98.6 ℉). Stress from extreme external temperature can cause the human body to shut down [...]

| SPLARE \| top features (document rank = 2) Explanation (from Neuronpedia) Lin (2023) | % | SPLADE \| top tokens (document rank = 73) Token | % |
|---|---|---|---|
| references to "human beings" and related concepts | 12.3 | ␣human | 29.3 |
| terms related to temperature variations and environmental conditions | 11.9 | ␣Human | 21.5 |
| terms related to fever and its physiological effects | 10.3 | ␣average | 20.7 |
| terms related to biological concepts and interactions | 9.6 | ␣humans | 20.3 |
| references to averages or average values in contexts related to statistics or metrics | 9.2 | ␣withstand | 3.8 |
| specific guidelines and recommendations related to health and wellness | 8.6 | ␣endurance | 2.4 |
| phrases related to summer and heat conditions | 8.2 | ␣limit | 2.1 |
| specific temperature values and their measurements | 6.8 | | |
| references to bodily systems and their components | 6.7 | | |
| quantitative data related to spending and financial metrics | 5.9 | | |

---

**Figure 15: Retrieval example from `CodeEditSearchRetrieval`**

**Query**: `Add totally untested pools ;)`
**Positive document**: `---`

```
+++
@@ -1,4 +1,6 @@
-import abc
+from multiprocessing import Pool as ProcessPool
+from multiprocessing.dummy import Pool as ThreadPool
+from multiprocessing import cpu_count
def do_flow(flow, result=None):
@@ -8,19 +10,41 @@
return result
+class PoolAPI(object):
+ def map(self, *args, **kw):
+ return self.pool.map(*args, **kw)
+
+
+class ThreadPool(PoolAPI):
+
+ de ...
```

| **SPLARE** \| **top features** (document rank = 10) | | **SPLADE** \| **top tokens** (document rank = 3) | |
|---|---|---|---|
| Explanation (from Neuronpedia) Lin (2023) | % | Token | % |
| references to "pool" and related concepts in various contexts | 32.4 | ␣pool | 17.0 |
| aspects of vacation experiences related to comfort and amenities | 28.0 | ␣pools | 15.1 |
| references to specific events or actions occurring in a timeline or sequence | 7.1 | ␣Pool | 14.5 |
| topics related to punk culture and its influence on music and community | 6.9 | pool | 8.9 |
| references to the term "stream" or its variations within contexts related to art and medi … | 5.8 | Pool | 8.8 |
| comments about changes and improvements, particularly in processes, products, or performa … | 5.5 | ␣pooling | 6.6 |
| references to swimming pools and related recreational facilities | 3.8 | ␣pool | 3.9 |
| references to programming tasks and contributions related to software development | 3.7 | ␣patch | 3.7 |
| references to programming interfaces and database management | 3.2 | ␣improvements | 3.1 |
| expressions of sports performance and competition | 1.4 | ␣patches | 2.4 |

---

**Figure 16: Retrieval example from `CodeEditSearchRetrieval`**

**Query**: `Make sure that the interests_register tables are created`
`Nose tries to run the interests_register tests, but they will`
`fail unless the interest_register app is added to INSTALLED_APPS,`
`because its ta ...`
**Positive document**: `---`

```
+++
@@ -8,7 +8,8 @@
'pombola.place_data',
'pombola.votematch',
'speeches',
- 'pombola.spinner' ) + \
+ 'pombola.spinner',
+ 'pombola.interests_register') + \
APPS_REQUIRED_BY_SPEECHES
# create the ENABLED_FEATURES hash that is used to toggle features on
and off.
```

| **SPLARE** \| **top features** (document rank = 14) | | **SPLADE** \| **top tokens** (document rank = 1) | |
|---|---|---|---|
| Explanation (from Neuronpedia) Lin (2023) | % | Token | % |
| keywords related to the concept of registration in various contexts | 9.5 | ␣apps | 5.7 |
| phrases relating to various aspects of "interest" in different contexts | 9.2 | ␣register | 5.3 |
| mentions of applications or software-related terminology | 5.9 | ␣register | 5.2 |
| features related to software libraries and their installation | 4.7 | interest | 5.1 |
| mentions of interest in various contexts or subjects | 4.0 | register | 5.0 |
| instances of the word "create" and its variations, indicating a focus on creation and gen … | 3.9 | ␣Interest | 4.8 |
| issues related to coding and technical errors in query parameters | 3.9 | ␣interests | 4.6 |
| phrases related to registration and enrollment | 3.0 | apps | 4.5 |
| elements related to importing and structuring code within modules | 2.8 | ␣interest | 3.7 |
| control keywords related to system configuration and management | 2.8 | ␣interest | 3.7 |

> **Figure 17: Retrieval example from `CodeEditSearchRetrieval`**
>
> **Query**: Update variables names in exam tests
> **Positive document**: ---
> ```
> +++
> @@ -17,16 +17,16 @@
> def test_create_biopsy_exam(self):
> from biopsy.models import Biopsy
> - specific_exam = create_specific_exam('Biopsy')
> + biopsy_exam = create_specific_exam('Biopsy')
> - specific_exam | should | be_kind_of(Biopsy)
> + biopsy_exam | should | be_kind_of(Biopsy)
> def test_create_necropsy_exam(self):
> from necropsy.mod ...
> ```
>
> | SPLARE \| top features (document rank = 19)
Explanation (from Neuronpedia) Lin (2023) | % | SPLADE \| top tokens (document rank = 1)
Token | % |
> |---|---|---|---|
> | terms associated with analysis and examination in a specialized medical context | 19.8 | ␣exam | 12.4 |
> | instances of the word "exam." | 17.2 | ␣tests | 9.0 |
> | occurrences of the word "test" and its variations in various contexts | 16.9 | ␣Exam | 8.9 |
> | references to exams and testing processes | 15.2 | ␣exams | 7.4 |
> | references to testing and test cases in programming contexts | 10.2 | exam | 6.4 |
> | references to unit testing and its associated concepts | 9.6 | ␣test | 6.2 |
> | references to notable figures or characters in a narrative context | 3.2 | Exam | 5.3 |
> | technical terms and keywords related to programming and computer science | 2.6 | ␣exam | 4.6 |
> | references to institutions or organizations in a structured context | 2.1 | ␣examination | 4.3 |
> | numerical values and measurements | 1.7 | tests | 4.0 |

