# OpenReview forum: "Learning Retrieval Models with Sparse Autoencoders"
_ICLR.cc/2026/Conference — ICLR 2026 Poster_

### Official Review · Reviewer_xNQK · 2025-10-28

**Soundness:** 3
**Presentation:** 4
**Contribution:** 3
**Rating:** 6
**Confidence:** 5

**Summary:**

This paper introduces SPLARE, a novel Learned Sparse Retrieval model that ingeniously integrates Sparse Autoencoders (SAEs) with existing LSR frameworks like SPLADE. SPLARE projects text representations into a "latent feature space" learned by a pre-trained SAE, which is designed to be more semantic and language-agnostic.Through extensive experiments, the authors demonstrate that SPLARE significantly outperforms its vocabulary-based counterparts on multilingual and out-of-domain retrieval tasks.

**Strengths:**

1.The paper originally combines the semantic feature decomposition of SAEs with learned sparse retrieval;
2.The experimental results are impressive. SPLARE consistently outperforms SPLADE-Llama in multilingual and out-of-domain settings and achieves a level of performance comparable to SOTA dense models on comprehensive benchmarks like MMTEB.
3.Excellent Generalization and Efficiency: SPLARE demonstrates strong multilingual generalization even when trained only on English data.

**Weaknesses:**

1. Lack of Detailed Cost Analysis: The introduction of the SAE module incurs additional computational and memory costs at inference time. While the paper mentions mitigating this by using intermediate LLM layers, it does not quantify the latency and memory overhead introduced by the SAE projection step itself.
2. Insufficient Context for Sparsity: The paper controls document vectors to ~400 non-zero dimensions. While this is sparse in a >130k-dimensional space, its advantage is not immediately obvious when compared to dense vectors, which may only have a few hundred to a thousand dimensions in total. The paper lacks a direct, end-to-end comparison of latency, index size, and memory usage between SPLARE (with inverted indexes) and a top-tier dense retriever (with an ANN index like HNSW) at a similar effectiveness level.

**Questions:**

1. Can you provide more information on the analysis and comparison of deployment costs?
2. Regarding sparsity control, the paper notes that combining training-time regularization with inference-time Top-K pruning was superior to using Top-K alone. Could the authors elaborate on the distinct effects of these two methods on the final representations? For example, does the regularization loss encourage the model to learn a better distribution of important features, which Top-K then effectively truncates, whereas relying solely on Top-K might crudely discard features that are contextually important but have marginally lower activation scores?
3. The model training relies on knowledge distillation from a cross-encoder. Was a contrastive learning approach, which is dominant in dense retriever training, considered or attempted?

---

> ### Author Response · Authors · 2025-11-20
>
> Dear reviewer, thank you for your feedback and for recognizing the originality as well as many benefits of our approach. We answer below to your comments and questions.
>
> > Lack of Detailed Cost Analysis: The introduction of the SAE module incurs additional computational and memory costs at inference time. While the paper mentions mitigating this by using intermediate LLM layers, it does not quantify the latency and memory overhead introduced by the SAE projection step itself.
>
> and
>
> > Can you provide more information on the analysis and comparison of deployment costs?
>
> We would like to clarify that introducing the SAE adds computational overhead relative to SPLADE only when the SAE width |W| exceeds the size of SPLADE’s vocabulary. In most of our comparisons, we use Llama Scope SAEs with |W|=131k, which is roughly equivalent to the Llama-3.1 8B vocabulary size; in this case, the cost (w.r.t. SPLADE) is essentially unchanged. Increasing the SAE width would indeed increase computation, but this overhead remains modest compared to the cost of several multi-head attention layers in the underlying model. In addition, as you point out, SPLARE models are usually trained at earlier layers which makes them more efficient by design.
>
>
> To further reinforce our point, we report (same answer provided to reviewer **7hiw**) the average forward inference time in ms, measured on synthetic queries of controlled length, batched in groups of 256, on a single A100 GPU. These results illustrate that (i) the projection head has a negligible impact on latency (SPLADE ≃ dense), (ii) SPLARE trained at layer 26 is slightly faster than SPLADE, and (iii) SPLARE trained at layer 6 is substantially faster than its layer-26 counterpart.
>
> | Model                                                               | q_l=16  | q_l=32  | q_l=64  | q_l=128 |
> |-|-|-|-|-|
> | Dense-Llama    | 1.72  | 2.89  | 5.30  | 10.27 |
> | SPLADE-Llama | 1.71  | 3.01  | 5.63  | 11.03 |
> | SPLARE (L=26) | 1.70  | 2.89  | 5.37  | 10.51 |
> | SPLARE (L=6) | 0.83  | 1.28  | 2.18  | 4.12  |

---

> > ### Author Response · Authors · 2025-11-20
> >
> > > Insufficient Context for Sparsity: The paper controls document vectors to ~400 non-zero dimensions. While this is sparse in a >130k-dimensional space, its advantage is not immediately obvious when compared to dense vectors, which may only have a few hundred to a thousand dimensions in total. The paper lacks a direct, end-to-end comparison of latency, index size, and memory usage between SPLARE (with inverted indexes) and a top-tier dense retriever (with an ANN index like HNSW) at a similar effectiveness level.
> >
> >
> > Your question echoes the remarks raised by reviewer **7hiw**. For completeness, we reproduce our response to that point below.
> >
> >
> > In general, measuring and comparing retrieval latency is highly challenging because different retrieval paradigms rely on different underlying assumptions (hardware and software). Dense retrieval with systems such as FAISS [1] typically presumes access to substantial compute resources—for example, multiple CPUs or even GPUs—to perform similarity search efficiently. Sparse retrieval methods based on inverted indexes are often implemented and evaluated as single-core systems able to deal with a large volume of queries in parallel [1], making direct latency comparisons nontrivial [3,4].
> >
> >
> > Nevertheless, we provide a comparison below, which already offers indicative evidence. As discussed in Section 5 (effectiveness-efficiency trade-off), SPLARE retrieval (Top-K = (40, 400)), evaluated with the Seismic library, takes roughly 5ms per query on a single-core machine for MS MARCO. For comparison, we also train a dense retrieval model based on Llama-3.1 8B (using the <EOS> token from the final layer) in our multilingual setting (Section 6, distillation on our 1.3M queries), using the sentence-transformers library. The resulting embeddings have dimension |d| = 4096. We then measure single-threaded query latency under the same conditions, using FAISS HNSW (with efSearch = 256) and `faiss.omp_set_num_threads(1)`; the chosen efSearch parameter led to a similar effectiveness level. All experiments are run on the same machine equipped with an AMD EPYC 7313 processor. We obtain:
> >
> > |                       |   MRR@10  | Latency (ms) |
> > |:--------------------------------|--------------------------:|-----------------------------------:|
> > |        SPLARE       |        39.7             |                               5.2 |
> > | Dense |                     39.4                |                                         4.5            |
> >
> > Overall, these results suggest that sparse and dense retrieval operate within a similar latency regime. A more comprehensive analysis—such as constructing full Pareto frontiers across library hyperparameters and benchmarking across multiple datasets—would be required for stronger conclusions. This comparison also does not capture additional considerations, such as the choice of optimal Top-K pooling for SPLARE or the use of matryoshka dimensionality reduction for dense embeddings. While these are important factors, we believe a thorough investigation is beyond the scope of this paper and is left for future work.
> >
> > **TL;DR**: Given the results discussed above (≈5 ms for inference on relatively short queries and ≈5 ms for retrieval), the end-to-end latency on a large-scale collection such as MS MARCO (8.8M passages) is approximately 10 ms.
> >
> > [1] Billion-scale similarity search with GPUs
> > [2] Efficient Inverted Indexes for Approximate Retrieval over Learned Sparse Representations
> > [3] Towards Effective and Efficient Sparse Neural Information Retrieval
> > [4] Moving Beyond Downstream Task Accuracy for Information Retrieval Benchmarking

---

> > > ### Author Response · Authors · 2025-11-20
> > >
> > > > Regarding sparsity control, the paper notes that combining training-time regularization with inference-time Top-K pruning was superior to using Top-K alone. Could the authors elaborate on the distinct effects of these two methods on the final representations? For example, does the regularization loss encourage the model to learn a better distribution of important features, which Top-K then effectively truncates, whereas relying solely on Top-K might crudely discard features that are contextually important but have marginally lower activation scores?
> > >
> > > These are good points, and it echoes the remarks raised by reviewer **EBaU**. For completeness, we reproduce our response to that point below.
> > >
> > > Although conceptually simpler, a top-K-pooled variant of SPLADE/SPLARE consistently yields models that lie below the Pareto frontier of effectiveness-efficiency when compared to training with a regularization loss + post cropping (initial experiments, not shown in the paper). This is an interesting question in its own right, but it is somewhat beyond the scope of this paper, as it concerns broader efficiency trade-offs for SPLADE-style models.
> > >
> > >
> > > Because regularization-based training is generally superior, we focus our experiments on that setting. However, this choice introduces a subtle challenge: it is difficult to predict in advance the exact sparsity level achieved by a given training run, and it makes comparison harder. For this reason, we adopt a “conservative’’ configuration with relatively light regularization, which allows flexible post-hoc pruning to any desired sparsity level. This approach is justified, as static pruning has been shown to be effective for SPLADE models [1]. Importantly, our conclusions remain unaffected by this design choice, since SPLARE and SPLADE—our main comparison—are trained and evaluated under identical conditions.
> > >
> > >
> > > We also suspect that using Top-K during training alters the optimization dynamics, since gradients propagate only through the selected top dimensions. In contrast, allowing the representations to remain “free” while applying a regularization term tends to yield a smoother and more stable regularization effect throughout training. Although this is an interesting observation, exploring these dynamics in depth would require a careful experimental study that is beyond the scope of this paper.
> > >
> > > [1] A Static Pruning Study on Sparse Neural Retrievers

---

### Official Review · Reviewer_EBaU · 2025-10-30

**Soundness:** 3
**Presentation:** 2
**Contribution:** 2
**Rating:** 6
**Confidence:** 5

**Summary:**

This paper proposes SPLARE (SParse LAtent REtrieval), a novel Learned Sparse Retrieval (LSR) approach that leverages pre-trained Sparse Autoencoders (SAEs) to represent queries and documents as sparse vectors over a latent feature space rather than the traditional vocabulary space. By inserting SAEs into intermediate layers of large language models (LLMs), SPLARE produces semantically rich, multilingual, and domain-generalizable sparse embeddings. The authors demonstrate through extensive experiments on benchmarks like MMTEB, MIRACL, and XTREME-UP that SPLARE consistently outperforms vocabulary-based LSR models—especially in multilingual and out-of-domain settings—while maintaining high retrieval efficiency. A 7B-parameter multilingual variant of SPLARE achieves competitive results against state-of-the-art dense retrievers, establishing it as the top-performing LSR model on MMTEB at the time of submission.

**Strengths:**

1.	The paper is the first to systematically use the pre-trained sparse autoencoder as the "implicit vocabulary" of LSR, replacing the traditional methods based on the original tokenizer vocabulary.
2.	The proposed method demonstrates outstanding performance in multilingual and cross-domain scenarios.

**Weaknesses:**

1.	Although the authors find that the best performance was achieved at Layer 26 for different SAE widths, this conclusion is only drawn from the experiments conducted on the Llama-3.1-8B model. If readers wish to apply the SPLARE method to other models (such as Gemma), there is no guarantee that the selection of this layer will be successful. It is suggested that the author provide more experimental results on the models and the layer selection strategies to enhance the generalization ability of the method.
2.	Current state-of-the-art dense retrievers generally adopt contrastive learning, but SPLARE chose distillation. The paper merely refers to distillation as a "common toolbox", but does not provide an explanation as to why distillation is more suitable for the sparse latent space. This crucial design choice lacks an analysis of motivation.
3.	The paper clearly states that only residual SAEs were used, but no ablation experiments or reasons are provided. SAEs using MLP or Attention might capture richer or different types of semantic information. Ignoring them could lead to information loss and limit the model's expressive power.
4.	Table 1 shows that in both the English and Multilingual benchmarks, SPLARE significantly lags behind SPLADE-Llama in the Code domain. The paper only mentions "the advantage diminishes" without providing any analysis (such as feature visualization, error cases). This may indicate that the SPLARE method has flaws when dealing with highly structured and symbolic text (such as code), and this needs to be emphasized.
5.	The abstract and introduction have repeatedly emphasized that SPLARE is the first LSR model that can match the SOTA dense model on MMTEB. However, Table 3 shows that its score of 60.9 is much lower than that of top dense models such as Qwen-3-Embedding-8B (70.9), inf-retriever-v1 (66.5), etc. The discussion in the paper contains an exaggeration.

**Questions:**

1.	The paper claims to propose "Learned Sparse Retrieval", but its core component SAE, is frozen throughout the training process. This means that the model cannot learn or optimize the "latent vocabulary" itself for retrieval purposes and can only be adapted to a fixed and non-designated feature space for the retrieval task through LoRA fine-tuning of the LLM's intermediate representations. This is fundamentally different from LSR methods such as SPLADE (whose LM head is learnable), and it weakens the claim of "Learned".
2.	The paper not only employs DF-FLOPS type sparse regularization but also adds Top-K pooling during inference to forcibly control the number of activations. The authors also admit that training solely with Top-K would be worse, but still choose the "moderately sparse" + post-cropping approach during training. This inconsistency will introduce non-differentiable post-processing and distribution drift, potentially leading to instability near the critical threshold and sorting reversals.

---

> ### Author Response · Authors · 2025-11-20
>
> Dear reviewer, thank you for your feedback and for recognizing the competitive aspects of SPLARE. We answer below to your comments and questions.
>
> > Although the authors find that the best performance was achieved at Layer 26 for different SAE widths, this conclusion is only drawn from the experiments conducted on the Llama-3.1-8B model. If readers wish to apply the SPLARE method to other models (such as Gemma), there is no guarantee that the selection of this layer will be successful. It is suggested that the author provide more experimental results on the models and the layer selection strategies to enhance the generalization ability of the method.
>
> Thank you for pointing that out. We have updated Fig. 2 (Left) to include an additional ablation using a Gemma Scope backbone (|W|=65k), and we observe the same overall trend. While the best performance often appears at relatively deep layers, this choice is not critical: performance at earlier layers remains strong, and the selection ultimately reflects an effectiveness-efficiency trade-off. A practical rule of thumb is to select a layer around two-thirds of the model depth to maximize **effectiveness**.
>
>
> However, as indicated in the general comment, we further complement our study by training a strong SPLARE model at layer L=6, following the same recipe as in our main experiments in Section 6. This yields a substantially more efficient model (approximately 2B parameters, with faster inference) that still achieves very competitive performance on MTEB Multilingual (57.6 average). We believe this model provides an appealing effectiveness-efficiency trade-off, and we will also include it in our public release.
>
> > Current state-of-the-art dense retrievers generally adopt contrastive learning, but SPLARE chose distillation. The paper merely refers to distillation as a "common toolbox", but does not provide an explanation as to why distillation is more suitable for the sparse latent space. This crucial design choice lacks an analysis of motivation.
>
> Thank you for this comment—this is indeed an insightful observation that we should have discussed more thoroughly.
>
>
> Since our retrieval model mirrors SPLADE (with the vocabulary projection being the only difference), we follow the established training practices for SPLADE. In particular, prior work has shown that distillation provides a highly effective training framework for SPLADE models, which motivates our reliance on it [1,2,3]. We also initially experimented with training using a contrastive loss, but this led to lower performance (for reasons discussed below).
>
>
> More broadly, your remark touches on an important point that extends beyond LSR. State-of-the-art embedding models typically rely on contrastive learning, which is generally effective but suffers from well-known issues such as false negatives. As a result, many recent systems incorporate filtering mechanisms for negative samples—often using cross-encoders or LLMs—which can themselves be viewed as implicit forms of distillation (for instance in Gemini Embedding [4]; this is also thoroughly studied in NV Retriever [5]).
>
>
> Ultimately, we choose distillation because (i) it has been shown to work particularly well for SPLADE models, and (ii) it offers a simple and direct way to mitigate some of the inherent limitations of contrastive learning—most notably the issue of false negatives, which are implicitly handled through the cross-encoder scores used during distillation.
>
>
> Nonetheless, to provide a concrete reference point, we trained a model using contrastive learning (without negative filtering) in the multilingual setting with a stable hyperparameter configuration (for Llama Scope, layer L=26). We then compared it to an equivalent model trained with distillation. Note that the performance of the distilled model differs from Table 3, as this comparison reflects a single training run rather than a merged model. We notice a 1.1 average performance drop; this behavior is consistent in various other scenarios we have explored, and is the main reason we rely on KLDiv. Note that additional filtering is likely to improve the contrastive results.
>
>
> |                       |   KLDiv | InfoNCE |
> |:--------------------------------|--------------------------:|-----------------------------------:|
> | MTEB(Multilingual, v2)               |                     60.3 |                               59.2 |
>
> [1] From Distillation to Hard Negative Sampling: Making Sparse Neural IR Models More Effective
> [2] SPLADE-v3: New baselines for SPLADE
> [3] A Unified Framework for Learned Sparse Retrieval
> [4] Gemini Embedding: Generalizable Embeddings from Gemini
> [5] NV-Retriever: Improving text embedding models with effective hard-negative mining

---

> > ### Author Response · Authors · 2025-11-20
> >
> > > The paper clearly states that only residual SAEs were used, but no ablation experiments or reasons are provided. SAEs using MLP or Attention might capture richer or different types of semantic information. Ignoring them could lead to information loss and limit the model's expressive power.
> >
> > This is a good remark. We focus on residual SAEs because they are trained directly on the output representations of each transformer layer. These representations can be viewed as intermediate token embeddings at earlier layers, making them closely analogous to the final token representations used in SPLADE. In addition, residual-stream SAEs are generally more standard in current practice and offer practical advantages: for example, the residual stream is smaller than the MLP activations, which makes SAE training and inference significantly more computationally efficient. That said, experimenting with attention and MLP SAEs is an interesting research direction left for future work.
> >
> > > Table 1 shows that in both the English and Multilingual benchmarks, SPLARE significantly lags behind SPLADE-Llama in the Code domain. The paper only mentions "the advantage diminishes" without providing any analysis (such as feature visualization, error cases). This may indicate that the SPLARE method has flaws when dealing with highly structured and symbolic text (such as code), and this needs to be emphasized.
> >
> > Thank you for this comment, it does indeed deserve additional discussion. We added in the paper (in Appendix G - Figures 15-17) examples of features contributions for SPLARE and token contributions for SPLADE, to better understand what’s going on. What’s striking is that the selected SPLARE features look fairly generic (e.g., “references to programming tasks and contributions related to software development”), and not as specific as the ones used for general domain (albeit multilingual) tasks. Our main hypothesis is that the SAEs from Llama Scope are not well suited for handling code data. An examination of the Llama Scope and Gemma Scope papers suggests that their training corpora contains relatively little code, which likely limits the models’ ability to capture code-specific latent features. SPLADE, however, can always fall back on vocabulary tokens, which—while not always the most suitable representation for retrieval in our view—tend to provide greater robustness in out-of-domain scenarios. A natural way to test this hypothesis would be to train domain-specific SAEs and evaluate their impact on performance; we leave this exploration to future work.
> >
> > > The abstract and introduction have repeatedly emphasized that SPLARE is the first LSR model that can match the SOTA dense model on MMTEB. However, Table 3 shows that its score of 60.9 is much lower than that of top dense models such as Qwen-3-Embedding-8B (70.9), inf-retriever-v1 (66.5), etc. The discussion in the paper contains an exaggeration.
> >
> > We agree that this point warrants clarification. Throughout the paper (including the abstract), we have been careful to state that SPLARE achieves “top” or “competitive” results on MTEB—intentionally using more nuanced wording than claiming strict state-of-the-art performance. Despite the existence of a common benchmark, comparing current approaches has become increasingly difficult, as they differ substantially in backbone models, pretraining objectives, fine-tuning data etc. In particular, recent systems such as Gemini Embeddings and Qwen-3 Embeddings have set a new performance bar on MTEB, surpassing previous methods by a substantial margin—though this comes at the cost of extensive pretraining and a strong reliance on large-scale synthetic data.
> >
> >
> > As noted in Section 6.2, SPLARE (which does not rely on any pretraining) remains ranked among the **top-10 models on MTEB (Multilingual, v2)** for retrieval, which justifies calling it a “top” model in a broad sense—even if the absolute performance gap with the very best models is non-trivial. It is also currently the best LSR model on MTEB.

---

> > > ### Author Response · Authors · 2025-11-20
> > >
> > > > The paper claims to propose "Learned Sparse Retrieval", but its core component SAE, is frozen throughout the training process. This means that the model cannot learn or optimize the "latent vocabulary" itself for retrieval purposes and can only be adapted to a fixed and non-designated feature space for the retrieval task through LoRA fine-tuning of the LLM's intermediate representations. This is fundamentally different from LSR methods such as SPLADE (whose LM head is learnable), and it weakens the claim of "Learned".
> > >
> > > This is a valuable point. In our current setup, the SAE is indeed frozen, meaning that we do not directly optimize the “latent vocabulary.” Our intuition—supported by our empirical observations—is that the features learned by high-quality SAEs are already well-suited for retrieval. Moreover, training SPLADE-style models is known to be challenging due to the instability of the optimization process: models frequently collapse (no positive activations) or diverge (rapid growth of the L0 norm). Freezing the projection layer substantially stabilizes training, which simplifies the experimental setup.
> > >
> > >
> > > While jointly optimizing the SAE could, in principle, offer additional flexibility and potentially improve effectiveness (for instance in the Code domain, following your previous remark), our initial attempts did not yield promising results. A more systematic exploration of this direction is needed. Finally, we note that SPLADE models typically fine-tune the LM head, but this is not a strict requirement. For example, prior work [1] has shown that freezing the LM head can achieve comparable performance. Finally, keeping the SAE frozen preserves the interpretability of its learned features, as illustrated in our examples.
> > >
> > > [1] Sparsifying Sparse Representations for Passage Retrieval by Top-k Masking
> > >
> > > > The paper not only employs DF-FLOPS type sparse regularization but also adds Top-K pooling during inference to forcibly control the number of activations. The authors also admit that training solely with Top-K would be worse, but still choose the "moderately sparse" + post-cropping approach during training. This inconsistency will introduce non-differentiable post-processing and distribution drift, potentially leading to instability near the critical threshold and sorting reversals.
> > >
> > > This is a valid point, and it reflects a deliberate experimental choice on our side. Although conceptually simpler, a top-K-pooled variant of SPLADE/SPLARE consistently yields models that lie below the Pareto frontier of effectiveness-efficiency when compared to training with a regularization loss + post cropping (initial experiments, not shown in the paper). This is an interesting question in its own right, but it is somewhat beyond the scope of this paper, as it concerns broader efficiency trade-offs for SPLADE-style models.
> > >
> > >
> > > Because regularization-based training is generally superior, we focus our experiments on that setting. However, this choice introduces a subtle challenge: it is difficult to predict in advance the exact sparsity level achieved by a given training run, and it makes comparison harder. For this reason, we adopt a “conservative’’ configuration with relatively light regularization, which allows flexible post-hoc pruning to any desired sparsity level. This approach is justified, as static pruning has been shown to be effective for SPLADE models [1]. Importantly, our conclusions remain unaffected by this design choice, since SPLARE and SPLADE—our main comparison—are trained and evaluated under identical conditions.
> > >
> > > [1] A Static Pruning Study on Sparse Neural Retrievers

---

> > > > ### Comment · Reviewer_EBaU · 2025-11-27
> > > >
> > > > Thank you for the detailed updates and new experiments. I appreciate the additional comparison on Gemma Scope. I have decided to keep my score

---

### Official Review · Reviewer_7hiw · 2025-11-01

**Soundness:** 3
**Presentation:** 2
**Contribution:** 2
**Rating:** 6
**Confidence:** 4

**Summary:**

This paper studies sparse auto encoders based latent features effectiveness in representing documents and queries. Authors claim that sparse autoencoders based learned sparse retrievers outperform their vocabulary projection based counterparts. The main contributions  of this paper are:
1. This paper introduces SPLARE a new sparse retrieval technique leveraging sparse auto encoders.
2. Detailed investigations of benefits of using latent vocabulary vs standard LLM vocabulary.
3. New 7B multilingual sparse retriever model.

**Strengths:**

The main strengths of the paper are as follows:
1. Idea of using latent features for document and query representation might have a lot of applications in text retrieval
2. The SPLARE learned sparse retrieval framework is interesting.
3. Detailed performance comparison with SOTA text retrieval models.

**Weaknesses:**

The main Weaknesses of the paper are as follows:
1.  Details about SPLARE working is missing. It might need some more explanation.
2. Terms like SAE width needs some more details.
3. Limited novels of the overall retrieval model when compared to sparse embed and other LSR techniques.
4. How SAEs are trained is not clear from the paper.
5. Paper writing have a lot of scope of improvement.
6. Limited performance improvement on MTEB retrieval benchmark when compared to SPLADE-LLama

**Questions:**

1. How does the proposed approach with sparse embed  paper https://research.google/pubs/sparseembed-learning-sparse-lexical-representations-with-contextual-embeddings-for-retrieval/?
2. Is availability of Pre-trained SAE a constraint of this model?
3. Any comparison of latency and training time when compared to spade and other dense models?

---

> ### Author Response · Authors · 2025-11-20
>
> Dear reviewer, thank you for recognizing the novelty, value and competitive aspects of SPLARE. We answer below to your comments and questions.
>
> > Details about SPLARE working is missing. It might need some more explanation.
>
> We would appreciate guidance on which elements you feel require further clarification. Pointing to the specific parts that may benefit from additional detail or explanation would help us address your concerns more effectively.
>
> > Terms like SAE width needs some more details.
>
> The SAE width is defined in Section 2.1 as the number of features in the latent space. This follows the standard terminology used in the SAE literature—for example, it is the same convention adopted in both the Gemma Scope and Llama Scope papers.
>
> > Limited novels of the overall retrieval model when compared to sparse embed and other LSR techniques.
>
> While SPLARE adopts an architecture similar to SPLADE and other LSR models, we argue that it introduces a meaningful shift in how sparse embeddings are constructed. Whereas most existing LSR approaches map queries and documents to (expanded) bags of words derived from the LM’s vocabulary, our method instead maps them to bags of **concepts** defined by the SAE. This represents a substantial departure from vocabulary-based models, yielding improved multilingual performance and opening the door to new applications—for example, multimodal extensions or domain-specific SAEs tailored to specialized retrieval tasks.
>
> > How SAEs are trained is not clear from the paper.
>
> and
>
> > Is availability of Pre-trained SAE a constraint of this model?
>
> In this work, we leverage high-quality, publicly available SAEs—specifically those released as part of the LlamaScope and GemmaScope suites—and focus our contribution on adapting these SAEs into a sparse retrieval model. Section 2.1 provides a brief introduction to SAEs, but their full training procedures are detailed in their respective papers. At a high level, these models are trained on large text corpora using a reconstruction loss combined with a sparsity-inducing mechanism (e.g., top-k sparsification in the case of LlamaScope).
> In addition, to demonstrate that our approach is easily extensible—and not reliant on pre-existing SAEs—we reproduce SPLARE results using an SAE that we train ourselves from scratch. Specifically, we train an SAE with the Sparsify library on layer 26 of Llama-3.1-8B, using a width of |W| = 131k (expansion factor x32). The SAE is trained with top-k=32 on the `EleutherAI/SmolLM2-135M-10B` dataset (≈10B tokens), which took roughly 2.5 days on 4×A100 GPUs. We then fine-tune a SPLARE model based on this SAE and compare it to the corresponding Llama Scope-based SPLARE model (English setting, as reported in Table 1). Overall, we are able to closely match the results; the small performance gap is likely attributable to differences in the SAE pretraining procedure, dataset size, or other training nuances, which could be explored further.
> |                  |   SPLARE (Llama Scope) | SPLARE (Sparsify) |
> |:---------------------------|---------------------------------------:| ---------------------------------------:|
> | ArguAna                    |       56.1  |                          54.7 |
> | CQADupstackGamingRetrieval |              56.9        |             58.2 |
> | CQADupstackUnixRetrieval   |            40.5       |                40.0   |
> | ClimateFEVERHardNegatives  |          19.4        |                  19.0   |
> | FEVERHardNegatives         |            79.0       |                 77.8 |
> | FiQA2018                   |              43.9               |      42.5 |
> | HotpotQAHardNegatives      |             67.0    |                  66.7 |
> | SCIDOCS                    |                 17.5           |       16.6 |
> | TRECCOVID                  |                85.2         |          83.3 |
> | Touche2020Retrieval.v3     |              63.5   |                  63.3 |
> | Average                    |                 52.9            |      52.2 |
>
> > Paper writing have a lot of scope of improvement.
>
> Could you please indicate which specific sections or aspects you find unclear or in need of improvement? This would help us revise the paper more effectively and address your concerns accurately.

---

> > ### Author Response · Authors · 2025-11-20
> >
> > > Limited performance improvement on MTEB retrieval benchmark when compared to SPLADE-LLama
> >
> > You raise an interesting point, and we agree that it deserves further discussion in the paper. If we focus on the multilingual-trained models (Section 6), it is correct that SPLARE and SPLADE-Llama appear close in overall performance on both the English and Multilingual tracks of MTEB (Table 1). For the English subset, this similarity is expected, as operating directly over the vocabulary may already capture most of the necessary signal for english-based retrieval.
> > However, in the multilingual setting, SPLARE shows a +0.6 average improvement, which we already consider meaningful. More importantly, if we isolate only the truly multilingual datasets (since some datasets in the multilingual benchmark are actually English-only), the gap becomes substantially larger, reaching +1.7 (as shown in the Table below). This suggests that SPLARE provides clearer gains precisely in the scenarios where multilingual generalization is required.
> > |                       |   SPLARE |   SPLADE |
> > |:--------------------------------|--------------------------:|-----------------------------------:|
> > | BelebeleRetrieval               |                      83.5 |                               82.4 |
> > | MIRACLRetrievalHardNegatives    |                      70.7 |                               68.8 |
> > | MLQARetrieval                   |                      83.2 |                               80.3 |
> > | StatcanDialogueDatasetRetrieval |                      36.7 |                               32.2 |
> > | TwitterHjerneRetrieval          |                      74.4 |                               75.3 |
> > | WikipediaRetrievalMultilingual  |                      90.9 |                               89.9 |
> > | Average                         |                      73.2 |                               71.5 |
> >
> > This is also confirmed by Table 2 (SPLARE outperforms SPLARE on all individual language datasets), as well as results on MIRACL (we have added the evaluation on MIRACL for SPLARE-Llama in Table 2 and Table 10 => +1.7 average performance) and XTREME-UP which further indicate the apparent benefit of latent features in a multilingual/cross-lingual scenario.
> >
> > > How does the proposed approach with sparse embed paper https://research.google/pubs/sparseembed-learning-sparse-lexical-representations-with-contextual-embeddings-for-retrieval/?
> >
> > We consider Sparse Embed to be somewhat orthogonal to the focus of our work. It introduces a hybrid—and substantially more complex—retrieval architecture that combines SPLADE-style sparse representations with ColBERT-style multi-vector encodings, which is conceptually unrelated to the latent concept representations produced by SAEs. It is also worth noting that models such as SPLADE-v3, largely due to their effective training pipeline, outperform Sparse Embed on both MS MARCO and BEIR while remaining conceptually simpler. This is the main reason we adopt SPLADE as our base architecture in this study.

---

> > > ### Author Response · Authors · 2025-11-20
> > >
> > > > Any comparison of latency and training time when compared to spade and other dense models?
> > >
> > > These are important points, thanks for catching it!
> > >
> > > **Regarding training time**: SPLARE and SPLADE-Llama are largely equivalent in architecture. Most of our experiments use LlamaScope, whose SAE has a width of |W|=131k—matching the size of the Llama vocabulary. When scaling the SAE width (e.g., |W| = 1M in Figure 2) with Gemma, the training time naturally increases to some extent.
> > > More importantly, we show that SPLARE can be trained at any transformer layer (Figure 2), provided a corresponding SAE is available. Our results indicate that intermediate layers generally yield the strongest performance. For example, our final model operates at layer 26 (out of 32), but we also demonstrate that highly competitive results can be obtained as early as layer 6 (see also our general remarks regarding the new contribution introduced with the 2B model). In this case, the training time is reduced substantially: for L=6, it is approximately divided by 2.5 compared to LLama-SPLADE or a dense model.
> > >
> > >
> > > **Regarding inference time**: In general, measuring and comparing retrieval latency is highly challenging because different retrieval paradigms rely on different underlying assumptions (hardware and software). Dense retrieval with systems such as FAISS [1] typically presumes access to substantial compute resources—for example, multiple CPUs or even GPUs—to perform similarity search efficiently. Sparse retrieval methods based on inverted indexes are often implemented and evaluated as single-core systems able to deal with a large volume of queries in parallel [1], making direct latency comparisons nontrivial [3,4].
> > >
> > >
> > > Nevertheless, we provide a comparison below, which already offers indicative evidence. As discussed in Section 5 (effectiveness-efficiency trade-off), SPLARE retrieval (Top-K = (40, 400)), evaluated with the Seismic library, takes roughly 5ms per query on a single-core machine for MS MARCO. For comparison, we also train a dense retrieval model based on Llama-3.1 8B (using the EOS token from the final layer) in our multilingual setting (Section 6, distillation on our 1.3M queries), using the sentence-transformers library. The resulting embeddings have dimension |d| = 4096. We then measure single-threaded query latency under the same conditions, using FAISS HNSW (with efSearch = 256) and `faiss.omp_set_num_threads(1)`; the chosen efSearch parameter led to a similar effectiveness level. All experiments are run on the same machine equipped with an AMD EPYC 7313 processor. We obtain:
> > >
> > > |                       |   MRR@10  | Latency (ms) |
> > > |:--------------------------------|--------------------------:|-----------------------------------:|
> > > |        SPLARE       |        39.7             |                               5.2 |
> > > | Dense-Llama |                     39.4                |                                         4.5            |
> > >
> > > Overall, these results suggest that sparse and dense retrieval operate within a similar latency regime. A more comprehensive analysis—such as constructing full Pareto frontiers across library hyperparameters and benchmarking across multiple datasets—would be required for stronger conclusions. This comparison also does not capture additional considerations, such as the choice of optimal Top-K pooling for SPLARE or the use of matryoshka dimensionality reduction for dense embeddings. While these are important factors, we believe a thorough investigation is beyond the scope of this paper and is left for future work.
> > >
> > >
> > > Additionally, we report the average forward inference time, measured on synthetic queries of controlled length, batched in groups of 256, on a single A100 GPU. These results illustrate that (i) the projection head has a negligible impact on latency (SPLADE ≃ dense), (ii) SPLARE trained at layer 26 is slightly faster than SPLADE, and (iii) SPLARE trained at layer 6 is substantially faster than its layer-26 counterpart.
> > >
> > > | Model                                                               | q_l=16  | q_l=32  | q_l=64  | q_l=128 |
> > > |--------------------------------------------------------------------------|-----|-----|-----|-----|
> > > | Dense-Llama    | 1.72  | 2.89  | 5.30  | 10.27 |
> > > | SPLADE-Llama | 1.71  | 3.01  | 5.63  | 11.03 |
> > > | SPLARE (L=26) | 1.70  | 2.89  | 5.37  | 10.51 |
> > > | SPLARE (L=6) | 0.83  | 1.28  | 2.18  | 4.12  |
> > >
> > >
> > > [1] Billion-scale similarity search with GPUs.
> > > [2] Efficient Inverted Indexes for Approximate Retrieval over Learned Sparse Representations.
> > > [3] Towards Effective and Efficient Sparse Neural Information Retrieval.
> > > [4] Moving Beyond Downstream Task Accuracy for Information Retrieval Benchmarking

---

### Author Response · Authors · 2025-11-20

We thank the reviewers for their thoughtful feedback, which has helped us clarify and improve several aspects of the paper. We believe that we have addressed all of the points raised in the reviews. Below, we first highlight a few general remarks, and then address each question in its respective review.

* We have made several minor revisions to the manuscript (with the main changes highlighted in green) to clarify and refine certain points. In addition, we have added new experiments and expanded our discussion, as detailed below.
* We updated Fig. 2 (Left) to complement the ablation with a Gemma Scope backbone. We also fixed the evaluation setting (for the same figure) with TopK=(40,400) (we were inadvertently not using pooling here). Conclusions remain similar.
* We evaluated SPLADE-LLama on MIRACL (Table 2 and Table 10), further providing evidence that SPLARE stands out in a multilingual setting (+1.7 average nDCG@10).
* As suggested by reviewer **EBaU**, we have added illustrative examples in the Code domain.
* We further complement our study by training a strong SPLARE model at layer L=6, following the same recipe as in our main experiments from Section 6 (multilingual training data with distillation, model merging from several runs, etc.). This results in a significantly more efficient model (approximately 2B parameters, with faster inference) that still achieves very competitive performance on MTEB Multilingual (57.6 average performance). We believe this model offers an attractive effectiveness–efficiency trade-off (while also highlighting the modularity of SPLARE), and we will include it in our public release.

---

### Author Response · Authors · 2025-11-26

Dear Reviewers,

Thank you again for your time and thoughtful feedback—we truly appreciate your efforts.

We hope that our rebuttal successfully addressed your questions and provided clearer evidence of the effectiveness of our approach. If there are still points that remain unclear, we would be grateful if you could share any updates or additional comments soon, as this would help ensure that we have sufficient time to respond before the discussion phase concludes.


If you feel that your concerns have been adequately addressed, we would also greatly appreciate an update to your review score.

---

### Author Response · Authors · 2025-12-03

Dear AC,

We would like to take this opportunity to summarize our work, emphasize several key points we consider essential, and consolidate our rebuttal.

# 1. Summary of our approach

* **SPLARE brings together two apparently separate research lines**—Sparse Autoencoders and Learned Sparse Retrieval—into a single unified framework. We believe this (i) represents an important shift in how sparse representations for retrieval can be constructed, (ii) opens new avenues for future research that should be of strong interest to the Information Retrieval community, and (iii) showcases a downstream application in which SAEs demonstrate particular effectiveness—a result we expect to be valuable for the mechanistic interpretability community as well.
* We show that the sparse latent features derived from two open-source SAE suites—Llama Scope and Gemma Scope—provide **more effective indexing units for documents and queries than the fixed LLM vocabulary**. This advantage is particularly pronounced in multilingual settings where we observe consistent and substantial improvements across benchmarks including MTEB Multilingual, XTREME-UP, and MIRACL.
* **Model contributions and positioning**: We release two SPLARE variants (2B and 7B) alongside our robust baseline SPLADE-Llama (8B), which already surpasses competitive systems such as Lion-SP-8B. **Collectively, these models form the first open multilingual suite of generalizable sparse embeddings extending beyond English, achieving performance competitive with state-of-the-art dense models across various MTEB benchmarks.**
* **Comparison to state-of-the-art models**: Although recent systems like Gemini Embeddings and Qwen3 Embeddings have set a new bar on MTEB through extensive pretraining and large-scale synthetic data generation, **SPLARE remains competitive without relying on either**. Its deliberately simple and transparent training setup—fine-tuning only on a large open-source dataset, without task-specific prefix instructions—ensures full reproducibility and leaves clear room for future gains through additional pretraining or data augmentation.

# 2. Summary of our rebuttal

* We have made several minor revisions to the manuscript (highlighted in green) to clarify specific points, and we have added new experiments together with an expanded discussion addressing the reviewers’ feedback.
* Figure 2 (Left) was updated to include a Gemma Scope backbone in the ablation (reviewer **EBaU**).
* We evaluated SPLADE-LLama on MIRACL (Tables 2 and 10), providing additional evidence of SPLARE’s strong multilingual performance (+1.7 average nDCG@10).
* Following reviewer **EBaU**’s suggestion, we added illustrative examples for the Code domain.
* We also trained a SPLARE model at layer 6 using the same multilingual distillation and model-merging pipeline as in Section 6. This ~2B-parameter model offers faster inference while maintaining competitive results on MTEB Multilingual (57.6 average).
* Clarifications: In our responses, we addressed several points raised by the reviewers, including our reliance on existing SAEs (reviewer **7hiw**), the choice of distillation over contrastive learning (reviewers **EBaU** and **xNQK**), or the effect of Top-K pooling (reviewers EBaU and xNQK).
* Efficiency (reviewers **7hiw** and **xNQK**): we provide evidence that sparse and dense retrieval systems operate within a comparable latency range.


In summary, we believe our rebuttal has effectively addressed all of the reviewers’ questions and concerns, resolving all the issues raised during the review process and providing a clearer and more complete picture of SPLARE’s capabilities. We sincerely thank the reviewers for their constructive feedback.

---

### Meta-Review · Area_Chair_vvJj · 2026-01-01

**Summary:**

All reviewers acknowledge the empirical contribution for outperforming SOTA methods or being comparable to them. The generalization and efficiency of the method.

There are concerns regarding the writing quality, the ablation study, and the rationale behind the design choices, which could potentially be addressed in the revision.

**Reviewer Concerns:**

Reviewer 7hiw:
- Details about SPLARE working is missing. It might need some more explanation. *unlikely to be solved*
- Terms like SAE width needs some more details. *solved*
- Limited novels of the overall retrieval model when compared to sparse embed and other LSR techniques. *partially solved*
- How SAEs are trained is not clear from the paper. *mostly solved*
- Paper writing have a lot of scope of improvement. *The question itself is vague and cannot be solved directly.*
- Limited performance improvement on MTEB retrieval benchmark when compared to SPLADE-LLama. *partially solved*
---
Reviewer EBaU:
- [Generalization capability]: Although the authors find that the best performance was achieved at Layer 26 for different SAE widths, this conclusion is only drawn from the experiments conducted on the Llama-3.1-8B model. If readers wish to apply the SPLARE method to other models (such as Gemma), there is no guarantee that the selection of this layer will be successful. It is suggested that the author provide more experimental results on the models and the layer selection strategies to enhance the generalization ability of the method.  *partially solved*
- [Design choice]: Current state-of-the-art dense retrievers generally adopt contrastive learning, but SPLARE chose distillation. The paper merely refers to distillation as a "common toolbox", but does not provide an explanation as to why distillation is more suitable for the sparse latent space. This crucial design choice lacks an analysis of motivation. *unlikely to be solved*
- [Ablation study] The paper clearly states that only residual SAEs were used, but no ablation experiments or reasons are provided. SAEs using MLP or Attention might capture richer or different types of semantic information. Ignoring them could lead to information loss and limit the model's expressive power. *partially solved; however, the reviewer (and readers) would like to see the comparison.*
- Table 1 shows that in both the English and Multilingual benchmarks, SPLARE significantly lags behind SPLADE-Llama in the Code domain. The paper only mentions "the advantage diminishes" without providing any analysis (such as feature visualization, error cases). This may indicate that the SPLARE method has flaws when dealing with highly structured and symbolic text (such as code), and this needs to be emphasized. *mostly solved*
- The abstract and introduction have repeatedly emphasized that SPLARE is the first LSR model that can match the SOTA dense model on MMTEB. However, Table 3 shows that its score of 60.9 is much lower than that of top dense models such as Qwen-3-Embedding-8B (70.9), inf-retriever-v1 (66.5), etc. The discussion in the paper contains an exaggeration. *mostly solved*
---
 Reviewer xNQK:
- Lack of Detailed Cost Analysis *mostly solved*
- Insufficient Context for Sparsity *mostly solved*

**Reviewer Scores:**

Reviewer 7hiw *is likely to keep the score*
Reviewer EBaU *has replied and keeps the score*
 Reviewer xNQK *is likely to keep the score*

---

### Decision · Program_Chairs · 2026-01-26

Accept (Poster)